# Single-cell analysis uncovers fibroblast heterogeneity and criteria for fibroblast and mural cell identification and discrimination

Lars Muhl [1,2✉], Guillem Genové [1,2], Stefanos Leptidis [1,2], Jianping Liu [1,2], Liqun He[3,4], Giuseppe Mocci[1,2], Ying Sun[4], Sonja Gustafsson[1,2], Byambajav Buyandelger[1,2], Indira V. Chivukula[1,2], Åsa Segerstolpe[1,2,5], Elisabeth Raschperger[1,2], Emil M. Hansson[1,2], Johan L. M. Björkegren [1,2,6], Xiao-Rong Peng[7], Michael Vanlandewijck[1,2,4], Urban Lendahl[1,8] & Christer Betsholtz [1,2,3✉]

Many important cell types in adult vertebrates have a mesenchymal origin, including fibroblasts and vascular mural cells. Although their biological importance is undisputed, the level of mesenchymal cell heterogeneity within and between organs, while appreciated, has not been analyzed in detail. Here, we compare single-cell transcriptional profiles of fibroblasts and vascular mural cells across four murine muscular organs: heart, skeletal muscle, intestine and bladder. We reveal gene expression signatures that demarcate fibroblasts from mural cells and provide molecular signatures for cell subtype identification. We observe striking inter- and intra-organ heterogeneity amongst the fibroblasts, primarily reflecting differences in the expression of extracellular matrix components. Fibroblast subtypes localize to discrete anatomical positions offering novel predictions about physiological function(s) and regulatory signaling circuits. Our data shed new light on the diversity of poorly defined classes of cells and provide a foundation for improved understanding of their roles in physiological and pathological processes.

[1] Karolinska Institutet/AstraZeneca Integrated Cardio Metabolic Centre, Blickagången 6, SE-14157 Huddinge, Sweden. [2] Department of Medicine Huddinge, Karolinska Institutet, SE-14157 Huddinge, Sweden. [3] Department of Neurosurgery, Tianjin Medical University General Hospital, Tianjin Neurological Institute, Key Laboratory of Post-Neuroinjury, Neuro-Repair and Regeneration in Central Nervous System, Ministry of Education and Tianjin City, Tianjin 300052, China. [4] Department of Immunology, Genetics and Pathology, Rudbeck Laboratory, Uppsala University, Dag Hammerskjölds väg 20, SE-75185 Uppsala, Sweden. [5] Klarman Cell Observatory, Broad Institute of MIT and Harvard, Cambridge, MA, USA. [6] Icahn Institute for Genomics and Multiscale Biology, Department of Genetics and Genomic Sciences, Icahn School of Medicine at Mount Sinai, One Gustave L. Levy Place, New York, NY, USA. [7] Bioscience Metabolism, Research and Early Development, Cardiovascular, Renal and Metabolism (CVRM) BioPharmaceuticals R&D, AstraZeneca, Gothenburg, Sweden. [8] Department of Cell and Molecular Biology, Karolinska Institutet, SE-17177 Stockholm, Sweden. ✉email: Lars.Muhl@ki.se; Christer.Betsholtz@ki.se

Fibroblasts, originally observed by Virchow[1] and Duvall[2] in the mid-1800s, are cells embedded within the fibrous or loose connective tissues of most mammalian organs[3]. Here, fibroblasts contribute to the formation and turnover of extracellular matrix (ECM), thereby providing tissues with different tensile properties and water content[4]. Fibroblasts also have important pathophysiological and pathological functions, e.g., in wound contraction and tissue fibrosis[5–7]. However, anonymous morphology and lack of specific molecular markers make resident tissue fibroblasts challenging to study[8] in contrast to other connective tissue cell types, e.g., osteoblasts/cytes, chondrocytes, adipocytes and blood cells. Fibroblast heterogeneity within and between organs therefore remains largely unexplored. It is also unclear how closely related fibroblasts are to vascular mural cells (vascular smooth muscle cells (SMC) and pericytes)[4,9–11] which, provide stability, contractility and elasticity to blood vessels and are dysregulated in several macrovascular and microvascular diseases[12,13].

To help bring more clarity to some of these questions, we here analyze fibroblasts, pericytes and SMC from four different organs in the mouse using single-cell RNA-sequencing (scRNAseq). The organs (heart, skeletal muscle, colon, and bladder) were chosen to harbor different types of muscle with or without mucosal linings, thereby encompassing physiological environments and demands that may reflect differences in fibroblast and mural cell functions. Our study provides molecular fingerprints for fibroblasts and mural cell archetypes and subtypes and reveals extensive inter- and intra-organ fibroblast heterogeneity in terms of molecular signatures and anatomical locations, implicating distinct physiological specializations. By decoding the complexity of some of the least known vascular and connective tissue cell types in the mammalian body, we provide molecular information important to advance our understanding of organ physiology and disease.

## Results

**Gene expression demarcates fibroblasts from mural cells.** Cells were collected from adult mouse heart, skeletal muscle, colon, and urinary bladder based on their expression of reporters and/or antibody-recognized epitopes for platelet-derived growth factor receptor-α (Pdgfra) and/or -β (Pdgfrb), chondroitin sulfate proteoglycan 4 (Cspg4) and smooth muscle actin α-2 (Acta2). These markers are known to be expressed broadly in mesenchymal cells including fibroblasts and mural cells[11,14,15]. Single-cell transcriptomes were generated from a total of 6158 cells using the SmartSeq2 protocol[16] (Fig. 1a, Supplementary Fig. 1a) and analyzed using the pagoda2 and SPIN algorithms[17–19], generating primary bar plot graphs and UMAP (uniform manifold approximation and projection)[20] visualization plots (Fig. 1b, Supplementary Fig. 1b, c).

Based on established markers for fibroblast and mural cell types in the brain vasculature[11,13], we first assigned preliminary annotations of fibroblast and mural cell archetypes in the combined dataset representing all four organs, and out of the 16 distinct clusters, 12 comprised fibroblasts, and 4 mural cells (Fig. 1c, Supplementary Data 1). Distinct fibroblast and mural cell clusters were also assigned in each organ separately (Fig. 1d, Supplementary Fig. 2a, b, Supplementary Data 1). UMAP visualization confirmed the pagoda2-SPIN clustering results, providing distinct separation of mural cells and fibroblasts in all organs (Supplementary Fig. 2c).

Next, a genome-wide list of fibroblast and mural cell-specific markers was produced for each organ by differential expression analysis. We used stringent criteria for gene qualification, including high expression level and specificity, in order to pinpoint markers suitable for immunohistochemistry and in situ

RNA hybridization analysis (Fig. 1e; Supplementary Datas 1–4; these and all following Supplementary data files are accessible at [https://betsholtzlab.org/Publications/FibroblastMural/database.html]). We searched the organ-specific lists for commonly expressed genes (Fig. 1f) to produce a short-list of putative universal markers distinguishing fibroblasts and mural cells. In this way, we found that 12.1% (45 out of 372) and 15.9% (45 out of 283) of the fibroblast- and mural cell-enriched genes, respectively, overlapped between all four organs (Fig. 1f). The short-list of common fibroblast markers included many ECM genes, such as Col1a1, Col1a2, Col5a1, Loxl1, Lum, Fbln1, and Fbln2, as well as the cell surface receptors Cd34 and Pdgfra. Importantly, this list did not contain the often-used marker fibroblast-specific protein-1 (FSP1 a.k.a. S100a4)[3,21]. The mural cell short-list encompassed the known mural cell markers Des, Mcam, Tagln and Notch3, but not the commonly used markers Pdgfrb, and Anpep (CD13)[13]. These data identify gene expression signatures that distinguish fibroblasts from mural cells across organs and pinpoint ambiguities with several commonly used markers. Of note, no single transcript qualified as a specific pan-fibroblast or pan-mural cell marker. For example, Pdgfra, which is missing in mural cells, is expressed in most, albeit not all, fibroblast subtypes (Fig. 1d). Likewise, Rgs5, which is missing in fibroblasts, is expressed in pericytes but missing in some SMC populations (Fig. 1d). The lack of universally specific markers distinguishing the fibroblast class of cells from the mural cell class of cells may not be entirely surprising considering their close relationship, but the relative specificities are nevertheless critical when selecting markers suitable for cell type discrimination by in situ expression analysis. In conclusion, molecular signatures composed of several markers, ideally chosen from the 90-gene short-list (Fig. 1f), are needed to robustly and universally distinguish between fibroblasts and mural cells. The complete lists of markers are available in Supplementary Table 1 and can be searched gene-by-gene at https://betsholtzlab.org/Publications/FibroblastMural/database.html.

**Fibroblast heterogeneity reflects distinct ECM profiles.** The dispersion of fibroblast clusters in the UMAP plots indicated a high degree of fibroblast heterogeneity (Fig. 2a, Supplementary Fig. 3a). This heterogeneity was primarily attributed to organ-specific differences (Fig. 2a), a conclusion supported by the pagoda2 cluster assignment (Fig. 2b). Overlaying the 16 pagoda2 clusters on top of the UMAP landscape showed a congruent cell type classification by the two methods (Fig. 2b). Since clustering of single-cell data may in part reflect batch- and cell injury-induced data skewing, it was important to couple clustering to distinct differences in marker gene expression. Indeed, the distribution of the top-50 marker genes for each of the 16 pagoda2 clusters identified fibroblast subtypes, reflecting organotypicity, as well as intra-organ heterogeneity (Supplementary Fig. 3b, Supplementary Data 5). In all analyses, the mural cells appeared considerably more homogeneous than the fibroblasts (Fig. 2a, b), despite the fact that mural cells include two cell types with clearly different cellular anatomies (pericytes and SMC).

To learn if any specific class of genes/proteins was a primary driver of the observed fibroblast heterogeneity, we investigated the contribution of different gene categories to the overall UMAP dispersion. These categories encompassed transcription factors[11,22] and genes associated with the Gene Ontology (GO) terms 'cytoskeleton', 'cell activation', 'cellular response to cytokine stimulus', 'cell surface receptor signaling pathway', 'cell–cell signaling', or 'ECM', the latter complemented with genes compiled in the Matrisome Project[23] (http://matrisomeproject.mit.edu) (Fig. 2c). Each gene category included 913–2603 genes,

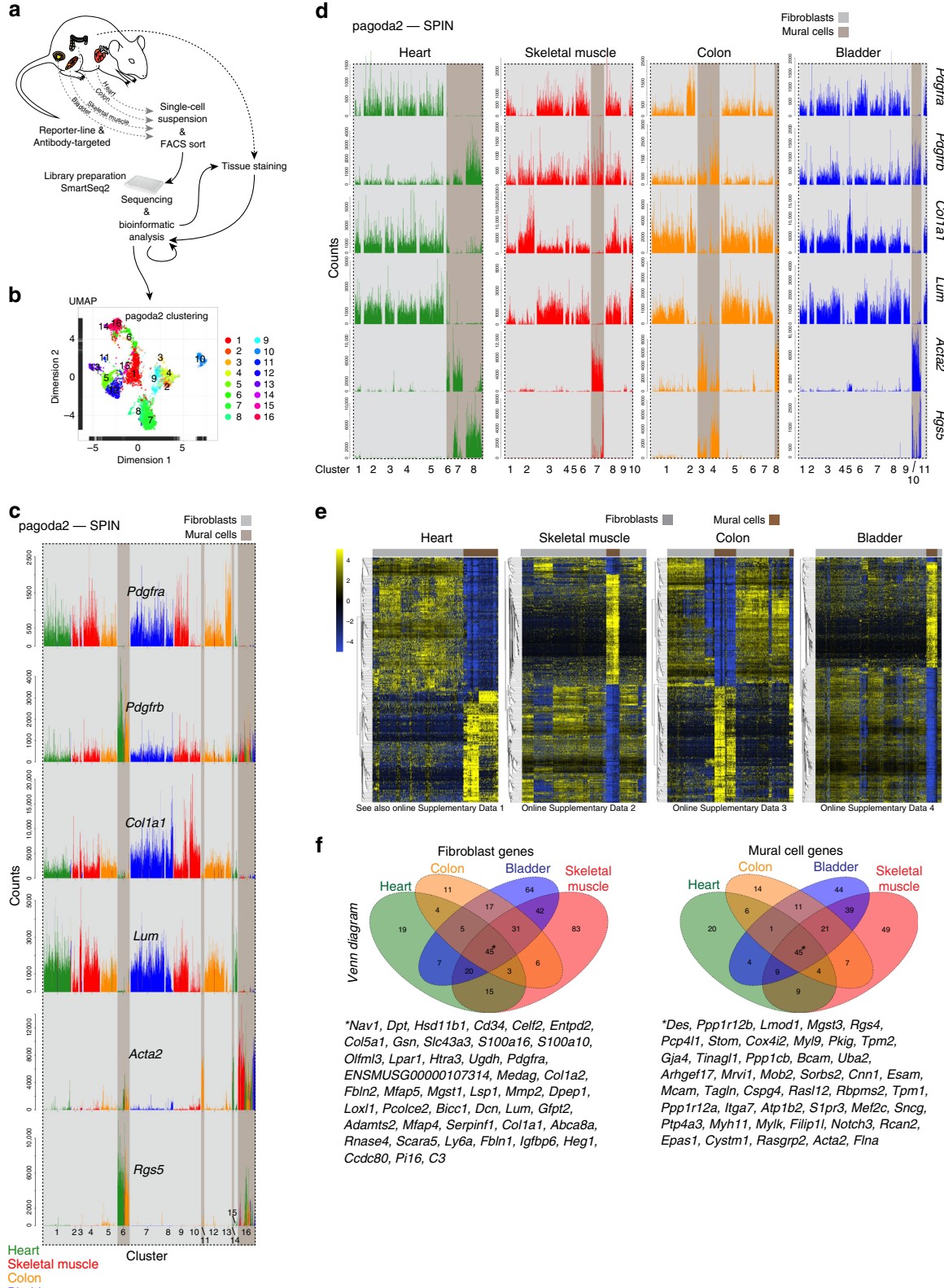

**Fig. 1 Outline and cell archetype identification. a** Study outline. **b** Pagoda2 clustering of the complete dataset (16 clusters) superimposed onto a UMAP dimensional reduction visualization. **c** Bar plots showing the expression of canonical fibroblast and mural cell markers in individual cells (bars) of the 16 pagoda2 clusters, the cell order in each cluster determined by SPIN. **d** Same data as in **c** but with pagoda2 clusters assigned and cells SPIN-ranged in each organ separately (see Supplementary Fig. 2a for corresponding UMAPs). **e** Expression heat maps (blue, low; yellow, high) showing the most differentially expressed genes between fibroblasts and mural cells in each respective organ dataset (for zoomable images, see Online Supplementary Datas 1–4). **f** Venn diagrams of differential expressed genes within the respective organ. Middle/asterisk: list of markers common for the four organs (see also Supplementary Table 1).

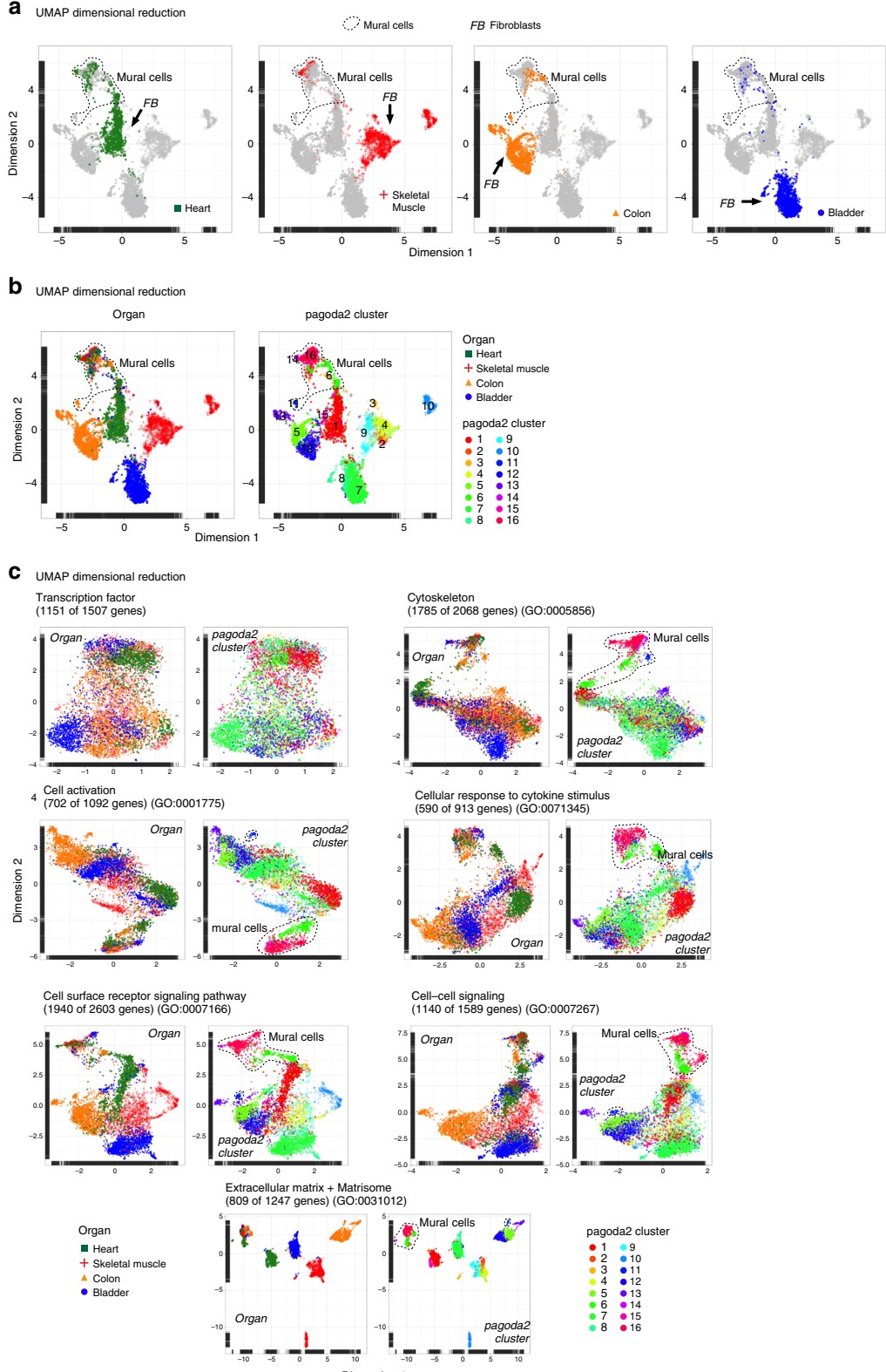

**Fig. 2 Cell dispersion criteria. a** UMAP visualization of the complete dataset, color coded for th respective organ of origin, separately. Cell clouds containing mural cells or fibroblasts are indicated. **b** UMAP visualization color coded for the organ of origin or pagoda2 clustering result. **c** UMAP visualization using restricted gene-sets color coded for the organ of origin combined, or pagoda2 clustering result. Cell clouds containing mural cells are indicated (see also Supplementary Data 2).

of which 590–1940 were expressed in our dataset (Supplementary Data 2). The ECM + matrisome gene-set stood out as the most potent driver of fibroblast heterogeneity, as indicated by the degree of dispersion in the UMAP landscapes (Fig. 2c, Supplementary Fig. 3c; the molecular details of these differences are provided in Supplementary Fig. 3d and Supplementary Data 6). The gene-sets for 'cell surface receptor signaling pathway' and 'cell-cell signaling' resulted in a cell dispersion similar to that obtained using all genes, whereas 'transcription factor', 'cellular response to cytokine stimulus' and 'cell activation' resulted in a more limited dispersion in UMAP visualization, suggesting that these gene categories were weaker contributors to the observed fibroblast heterogeneity (Fig. 2c). The 'cytoskeleton' gene-set separated the SMC and a subset of the pericytes away from the rest of the cells. These cells were α-smooth muscle actin (αSMA, Acta2) positive, suggesting, as expected, that a unique cytoskeletal composition is associated with cells endowed with certain contractile properties (Fig. 2c, Supplementary Fig. 3e). In conclusion, the strong fibroblast diversity primarily reflects differences in ECM production and maintenance, suggesting that fibroblasts tailor ECM production in an organ- and location-specific manner.

**Fibroblast subtypes localize to distinct anatomical niches**. The single-cell fibroblast transcriptomes indicated not only inter-organ, but also intra-organ heterogeneity, again with the ECM/matrisome as major driver (Fig. 2a–c). An important question was therefore whether these fibroblast subtypes occupy distinct anatomical niches within each organ where they contribute niche-specific ECM, or whether they contribute complementary ECM components to the same niche(s). To address this question, we localized fibroblast subtypes within their respective organ.

**Skeletal muscle**. Skeletal muscle harbors a complex arrangement of connective tissue and ECM[24,25]. The endomysium surrounds individual muscle cells (fibers), the perimysium surrounds groups of fibers (fascicles) and the epimysium surrounds entire muscle heads (Fig. 3a). The perimysium is continuous with the tendons[26] and exhibits a higher collagen type-I to type-III ratio, compared to the endo- and epimysium[25]. However, it is not well understood whether different fibroblast subtypes reside in these different tissue layers, and if so, what ECM and proteoglycan components they produce[27,28].

Two of the skeletal muscle fibroblast clusters displayed high expression of thrombospondins (THBS)-1 and -4, and type-XI collagen α-1 ($Thbs1^+$ $Thbs4^+$ $Col11a1^+$) (Fig. 3b) combined with low expression of type-III collagen α-1 (Col3a1) and Pdgfra (Supplementary Fig. 4a), suggesting that these clusters represent perimysial cells. THBS4-immunofluorescence localized these cells primarily to fasciae structures (Fig. 3c), confirming their perimysial identity and previous results regarding THBS4 expression in skeletal muscle[29]. Perimysial cells express several genes associated with tendon and cartilage development, e.g., Col12a1, connective tissue growth factor (Ctgf), Col11a2, tenascin-C (Tnc), scleraxis (Scx), and tenomodulin (Tnmd) (Fig. 3b, d, Supplementary Fig. 4a). It has been proposed that the perimysium is heterogeneous[24]. Accordingly, we observed molecular diversity within the perimysial cell clusters, exemplified by the expression of Wif1 (an inhibitor in the WNT pathway), Col22a1 (a collagen suggested to be present at the myotendinous junction and important for its stabilization)[30], Chodl (chondrolectin) and Rflnb (refilin B) in distinct sets of perimysial cells (Fig. 3e, f, Supplementary Fig. 4b) and differential expression of a large number of matrisome as well as non-matrisome genes across a SPIN range of the perimysial cells (Fig. 3f,

Supplementary Data 7). To what extent this heterogeneity reflects different anatomical location of different perimysial cell subtypes/states remains to be investigated. In addition to the perimysial cells, we identified a second $Thbs4^+$ skeletal muscle fibroblast subtype (pagoda2 cluster 4), which was $Thbs1^{low}$ $Thbs4^+$ $Col11a1^{low}$ and further expressed Pdgfra and periostin (Postn) (Fig. 3b, Supplementary Fig. 4a). POSTN immunofluorescence localized these cells to the interface between the perimysium and the endomysium (Fig. 3g), and because of this unique localization and the distinct gene expression profile we refer to these cells as paramysial cells. In experiments where m. soleus and m. gastrocnemius were processed separately, paramysial cells (marked also by C1qtnf3 and Cthrc1) were more abundantly captured from M. soleus (Fig. 3h) suggesting that fibroblast subtype abundance may differ between muscles. Although paramysial cells co-expressed several genes with perimysial cells (Thbs4, Col12a1, C1qtnf3, and Cthrc1) their distribution in UMAP was closer to the endomysial cells (Fig. 3b, h), which formed five pagoda2 clusters (#1, 3, 5, 6, 8) with limited dispersion in the UMAP landscape (Supplementary Fig. 2a). We anticipate that these clusters harbor fibroadipogenic progenitors and perivascular fibroblasts in addition to endomysial fibroblasts, but this will require further analysis. Finally, we identified an additional fibroblast subtype in the skeletal muscle (pagoda2 cluster 10) defined by expression of Cspg4, Pdgfra, and nerve growth factor receptor (Ngfr) (Supplementary Fig. 2a, Supplementary Fig. 4a, c). These cells were localized at the margin of nerve fibers that are often found close to major blood vessels (Supplementary Fig. 4c). Collectively, these data reveal specific fibroblast subtypes in skeletal muscle, with distinct anatomical locations.

**Heart**. The heart lacks perimysial ECM layers (Fig. 4a). Nevertheless, we found a cardiac fibroblast subtype (pagoda2 cluster 1) with distinct transcriptional similarity to skeletal muscle perimysial cells, including shared expression of Wif1, and cartilage oligomeric matrix protein (Comp) (Supplementary Fig. 5a). WIF1 and CYP2E1 immunofluorescence and Wif1 RNAscope localized these cells to the cardiac valves and their adjacent hinge regions (Fig. 4b–d, Supplementary Fig. 5b). These fibroblasts are likely identical to one or more of the recently described cardiac valve interstitial cell types[31,32]. We found ten commonly enriched genes in skeletal muscle perimysial and cardiac valve interstitial cells, including Thbs1, Comp, and Fmod (fibromodulin) (Fig. 4e, f, Supplementary Table 2), similarities that may reflect common functions related to ECM tensile strength. Similar to the skeletal muscle endomysial cells, the majority of the cardiac fibroblasts distributed into four pagoda2 clusters (# 2–5) with limited dispersion in the UMAP landscape (Supplementary Fig. 2a). Putative heterogeneity within this major cardiac fibroblast population and its similarity to skeletal muscle endomysial and perivascular fibroblasts awaits further investigation.

**Colon**. The colonic mucosa lacks villi but harbors millions of regularly spaced crypts (Fig. 5a). Because molecular heterogeneity along a mesenchymal crypt–villus axis has been demonstrated in small intestinal villi[33–35], we asked if a similar mesenchymal crypt–surface axis (from base to apex) could be defined within the colonic mucosa. Indeed, we found that distinct subpopulations of colon fibroblasts differed in terms of gene expression (Fig. 5b) and localization along the crypt–surface axis. Cells expressing Tnc, but not Cd34 ($Tnc^+$ $Cd34^-$) were localized immediately beneath the surface epithelium (Fig. 5c, d, Supplementary Fig. 6a). In contrast, $Tnc^-$ $Cd34^+$ fibroblasts were located deeper down in the lamina propria and in the muscularis mucosa (the

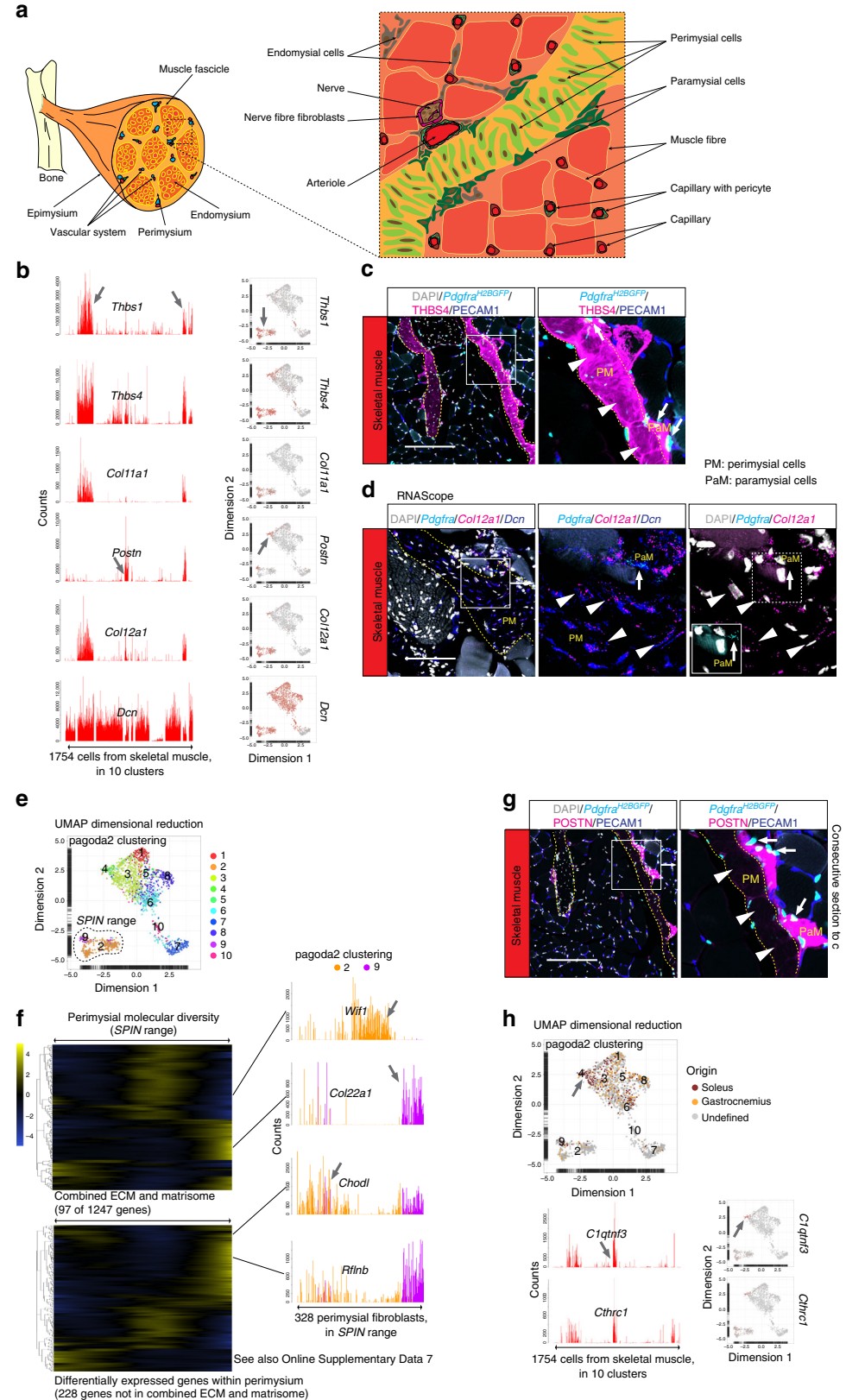

*Tnc* expression in the muscularis externa likely originates from SMC in this layer) (Fig. 5e, f). The *Tnc⁻ Cd34⁺* fibroblast population is probably similar to the recently described crypt stromal cells that are implicated in epithelial stem cell maintenance[36], as well as to previously described 'WNT ligand-secreting mesenchymal' cells[37,38]. The localization of *Tnc⁺*

*Cd34⁻* fibroblasts at the apex of the crypt was confirmed by interleukin-33 receptor (IL33R, encoded by *Il1rl1*) immuno-fluorescence co-localizing with *Pdgfra⁺* and COX-2⁺ (*Ptgs2*) (Supplementary Fig. 6b) cells, which have previously been shown to reside at the apex of the crypt[39]. These two major fibroblast populations also differed with regard to expression of

**Fig. 3 Fibroblast subtypes of the skeletal muscle. a** Schematic depiction of skeletal muscle anatomy. **b** Bar plots and UMAP visualization (gray, low; red, high expression) showing examples of genes with cell subtype-specific expression (arrow). **c** Immunofluorescence staining of skeletal muscle from *Pdgfra*$^{H2BGFP}$ reporter line for THBS4 and PECAM1. **d** RNAscope staining for *Pdgfra*, *Col12a1*, and *Dcn*. **e** UMAP visualization with pagoda2 clusters annotated and indication of cells specifically analyzed with *SPIN*. **f** Expression heat map (loess smoothed (locally weighted scatterplot smoothing) values; blue, low; yellow high) of 97 genes in the ECM + matrisome gene-set (upper) and 228 additional genes (lower) of the perimysial cells SPIN range (see also Online Supplementary Data 7). Bar plots showing examples of perimysial differentially expressed genes (arrows; *Wif1*, *Col22a1*, *Chodl*, and *Rflnb*) color coded based on pagoda2 clustering (# 2, 9). **g** Immunofluorescence staining of skeletal muscle from *Pdgfra*$^{H2BGFP}$ reporter line for POSTN and PECAM1 (consecutive section to **c**). Arrowheads: perimysial cells (PM); arrows: paramysial cells (PaM). **h** UMAP visualization, color coded for cellular origin according to muscle subtype (*M. soleus*, *M. gastrocnemius* or undefined), and pagoda2 clusters annotated. Arrow indicates pagoda2 cluster 4, which is enriched in cells specifically captured from soleus muscle (upper panel). Bar plots and UMAP showing examples of cluster four enriched genes (arrows; *C1qtnf3* and *Cthrc1*). Scale bar: **c**, **g** 200 μm, **d** 100 μm.

components of the BMP and WNT signaling pathways, which are important for epithelial differentiation in the colon[40,41]. The apex *Tnc*+ *Cd34*– fibroblasts specifically expressed BMP ligands (*Bmp2*, *Bmp5*, and *Bmp7*), whereas *Tnc*- *Cd34*+ fibroblasts at the crypt base expressed WNT ligands (*Wnt2* and *Wnt2b*) and receptors; frizzled class receptor 1 (*Fzd1*), *Fzd4* and the secreted soluble frizzled related protein 1 (*Sfrp1*) (Supplementary Fig. 7a, b, Supplementary Data 8). Interestingly, the forkhead box L1 (*Foxl1*) transcription factor, previously described to play important roles in intestinal development[42], was amongst the crypt apex fibroblast-enriched genes, which was confirmed by RNA-scope in situ RNA hybridization (Supplementary Fig. 7b, c). In sum, these observations show that fibroblasts contribute to a mesenchymal crypt-surface axis in colon, a mesenchymal differentiation that is likely of critical importance for the epithelial homeostasis[43].

**Bladder.** The bladder has an anatomical configuration similar to that of the colon in the sense that epithelial (urothelial) cell layers reside on top of a mesenchyme-containing mucosa, together forming the inner lining of the detrusor muscularis[44,45] (Fig. 6a). As in the colon, the bladder mucosa also contained fibroblast subtypes defined by *Tnc*+ *Cd34*– and *Tnc*– *Cd34*+ signatures (Supplementary Fig. 8a). The *Tnc*+ *Cd34*– fibroblasts were located immediately subjacent to the urothelium, whereas *Tnc*– *Cd34*+ fibroblasts resided at deeper locations in the bladder mucosa (Fig. 6b–d, Supplementary Fig. 8b, c). The colon and bladder fibroblast subtypes displayed additional cross-organ similarities; 13 commonly enriched genes were found in colon and bladder *Tnc*+ *Cd34*– fibroblasts, including *Ptgs1*, *Bmp5*, *Bmp7*, and *Cxcl14* (Fig. 6e, f, Supplementary Fig. 8c, Supplementary Table 3). We also found that *Col8a1* exhibited a similar expression pattern in colon and bladder muscularis and mucosal regions (Supplementary Fig. 8d). Despite these similarities, the colon and bladder fibroblast populations showed organotypic features. For example, bladder but not colon *Tnc*+ *Cd34*– fibroblasts expressed *Acta2*, possibly indicating differences in physiological contractility (Supplementary Fig. 8c). Furthermore, the sub-urothelial *Tnc*+ *Cd34*– fibroblasts expressed *Bmp2* and *Bmp3* in addition to *Bmp5* and *Bmp7* (Supplementary Fig. 8c). Together, these data reveal both important similarities and organotypic differences between colon and bladder fibroblasts.

**Mural cells show less heterogeneity than fibroblasts.** Mural cells were substantially less heterogeneous than fibroblasts, both within and across the four analyzed organs (Figs. 1c, d and 2a). Pagoda2 clustering of all cells separated the mural cells into four distinct clusters (# 6, 11, 14 and 16 in Fig. 1b, c), and further individual analysis of the mural cells alone separated them into eight clusters (Fig. 7a). The spatial distribution in the UMAP visualization indicated a gradual phenotypic transition from pericytes to SMC,

exemplified by the expression of the marker genes *Acta2*, *Myh11*, *Rgs5*, and *Kcnj8*[46,47] (Fig. 7a, Supplementary Fig. 9a, Supplementary Data 1). Minor organ-specific clustering or spatial dispersion was observed, especially amongst the *Acta2*+ SMC (Fig. 7a, Supplementary Fig. 9a). The phenotypic continuum between pericytes and SMC indicated by pagoda2 clustering and UMAP visualization was corroborated by trajectory analysis using monocle (Fig. 7b). This analysis revealed one major branch for pericytes, one possible intermediate branch, and two SMC branches (Fig. 7b, Supplementary Fig. 9b). Similar to UMAP, the trajectory analysis indicated a high degree of cross-organ mural cell homogeneity. Heat map analysis of the top 50 marker genes for each cluster supported the similarities between the pericyte (# 1, 3) and SMC (# 2, 4, 5, 8) clusters, while clusters 6 and 7, representing mural cells originating from the colon or heart, respectively, appeared more distinct (Fig. 7c, Supplementary Data 9), which is in agreement with their separate distribution in UMAP (Fig. 7a). We found *Hhip* and *Rgs10* specifically expressed in colon cells in cluster 6 (Supplementary Fig. 9c). HHIP immunohistochemistry revealed that these cells, which are *Acta2* positive but *Cspg4* and *Pdgfra* negative, correspond to SMC located inside the colonic mucosal layer (Supplementary Fig. 10a). These mucosal SMC are not associated with blood or lymphatic vessels, but instead stretch from the mucosal base to its surface, similar to the lacteal-associated SMC located in the small intestinal villi[48] (Supplementary Fig. 10a). Finally, the heart mural cells in cluster 7 are *Sost*+ *Cbr2*+ (Supplementary Fig. 9c) and related to SMC from the lung, as discussed below.

*Pdgfrb*$^{GFP}$-high cells in all four organs displayed a morphology and capillary association typical for pericytes (Fig. 7d). These cells were clearly visible and abundant in all organs, consistent with a ubiquitous presence of pericytes in capillaries, while their capture rate for scRNAseq analysis differed markedly between organs, ranging from numerous pericytes from heart and colon to only a few (≈10 cells) from skeletal muscle or bladder. Possible explanations for these differences include that the pericytes of certain organs are either more sensitive and die during tissue dissociation, that they are more firmly adhered to the endothelial cells and get excluded by counter-selection of endothelial cell (fragment) contamination during cell isolation, or that they are similar to and thereby end up clustering together with venous SMCs (e.g., in bladder—see below).

We also noticed organ-specific differences in pericyte localization; a subset of colon pericytes was invariably located at the far side of the subepithelial capillary loops relative to the surface epithelium (Fig. 7e, Supplementary Fig. 10b, Supplementary Movies 1 and 2). These pericytes were strongly positive for *Pdgfrb*$^{GFP+}$ and *Cspg4*$^{dsRED+}$, but weak for SMC markers αSMA and CNN1 (Fig. 7e, f). In contrast, many bladder pericytes expressed *Acta2*$^{GFP}$ and αSMA (Supplementary Fig. 10c, d) which may relate to vascular anatomy: the diameter of bladder mucosal capillaries was substantially wider

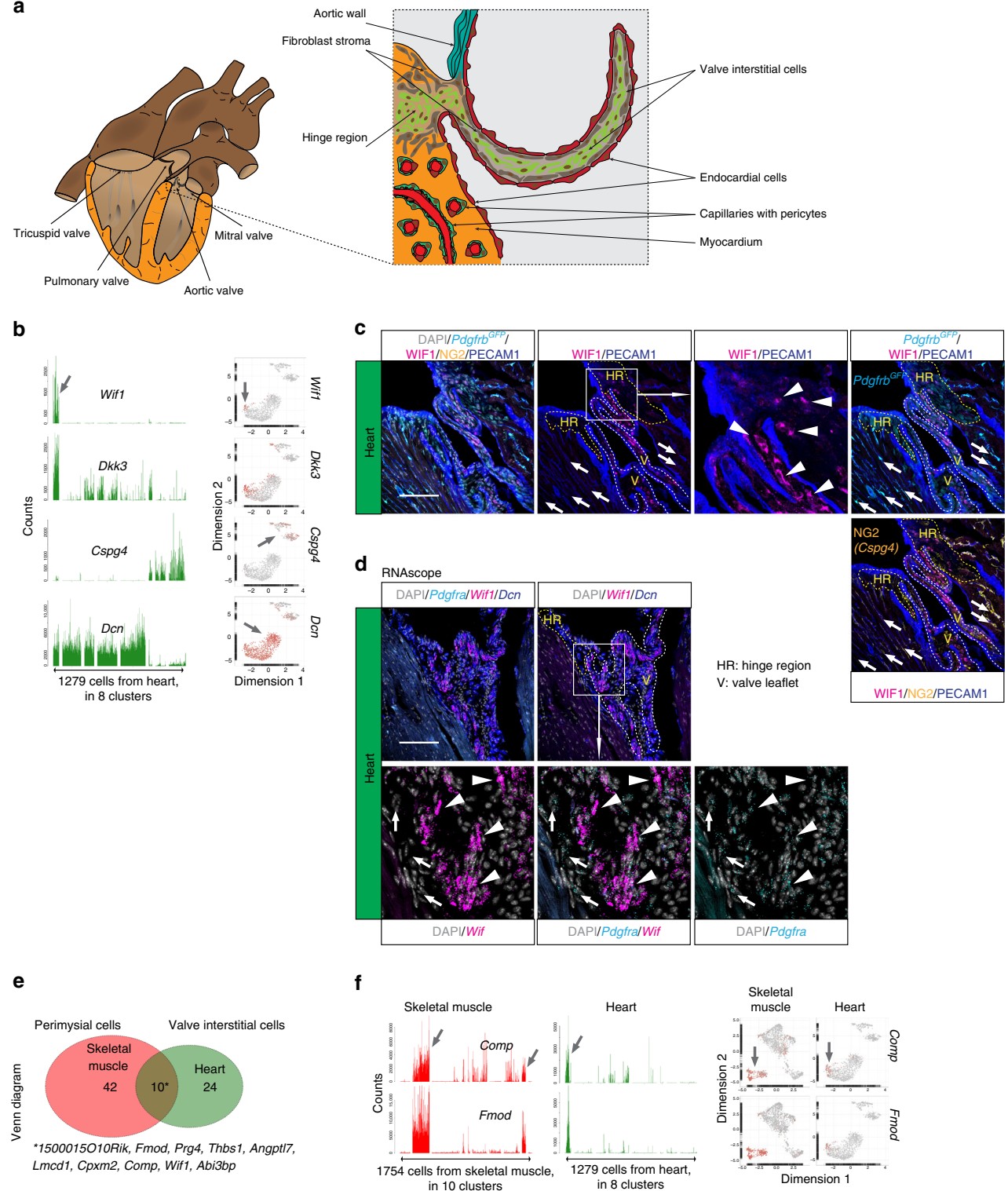

**Fig. 4 Fibroblast subtypes of the heart. a** Schematic depiction of heart anatomy. **b** Bar plots and UMAP visualization (gray, low; red, high expression) showing examples of cell subtype-specific expression (arrow). **c** Immunofluorescence staining of heart from the *Pdgfrb*^GFP reporter line for WIF1, NG2, and PECAM1, focused on the cardiac valve and hinge region. **d** RNAscope staining for *Pdgfra*, *Wif1*, and *Dcn*, focused on the cardiac valve region. Arrowheads: valve interstitial cells; arrows: pericytes; HR: hinge region, V: valve leaflet. Scale bar: 100 μm. **e** Venn diagram showing skeletal muscle perimysial- and heart valve interstitial cell co-enriched genes. **f** Examples of co-enriched genes presented as bar plots and UMAP (gray, low; red, high expression). Asterisk: gene list of commonly enriched genes (see also Supplementary Table 2).

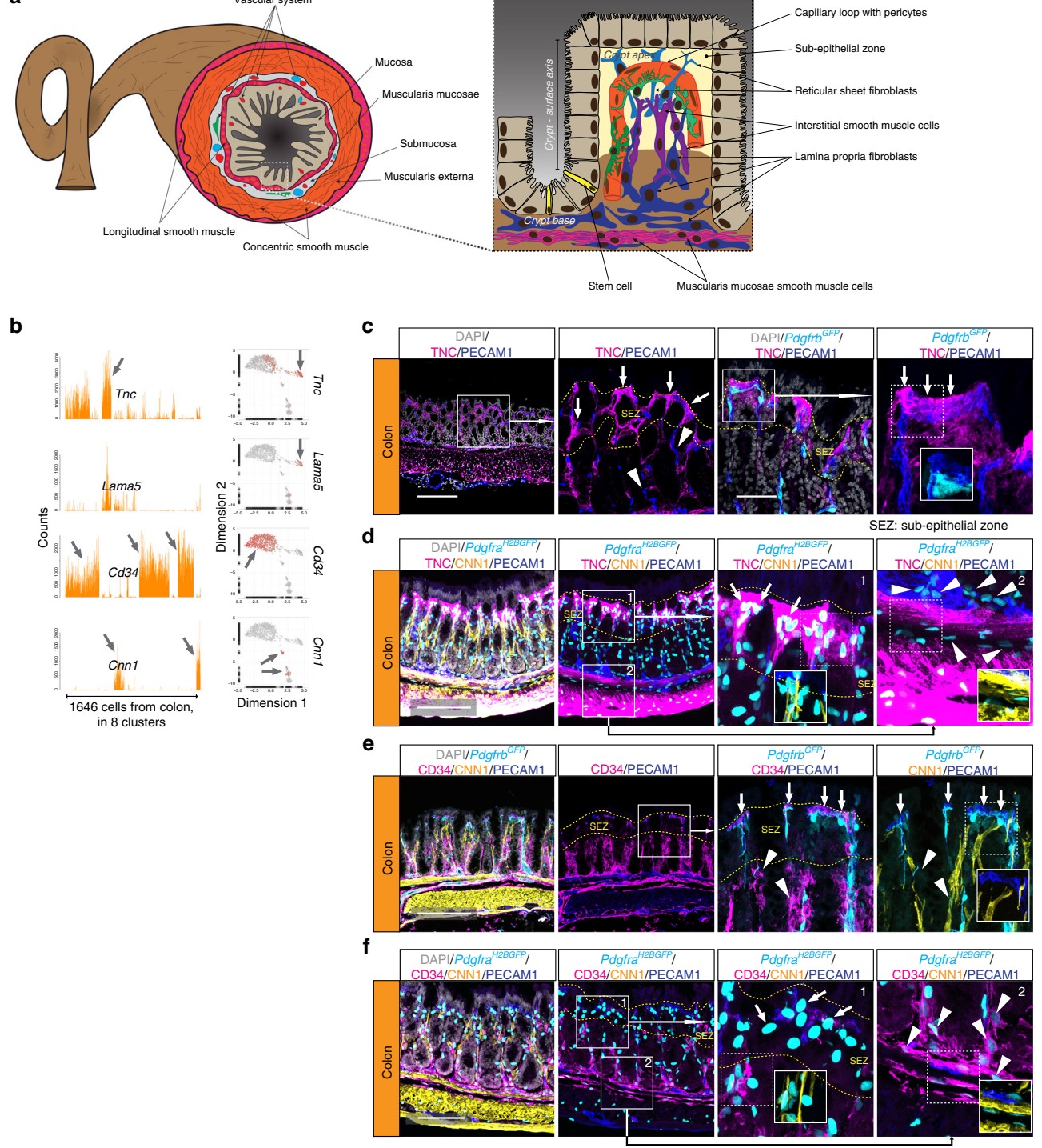

**Fig. 5 Fibroblast subtypes in the colon. a** Schematic depiction of colon anatomy. **b** Bar plots and UMAP visualization (gray, low; red, high expression) showing examples of genes with cell subtype-specific expression (arrow). **c–f** Immunofluorescence staining of colon samples from reporter lines *Pdgfrb*GFP (**c, e**) and *Pdgfra*H2bGFP (**d, f**) for indicated markers; **c, d** TNC+ cells close to the crypt apex surface (arrows); **e, f** CD34+ cells at the crypt base (arrowheads); **d, f** *Pdgfra*+ cells close to the crypt apex surface, negative for CD34 or CNN1 (arrows). Arrowheads: *Pdgfra*+ CD34+ cells at crypt base and muscularis mucosae, negative for CNN1. Scale bar: **c, e** 200 μm, **d, f** 100 μm.

(6.8 ± 1.1 μm) than the capillary diameter in for example heart, skeletal muscle, or colon (4.3 ± 0.2 μm) (Supplementary Fig. 10d, e), correlating with the venous SMC-like appearance of these bladder pericytes[11]. Taken together, our data show that pericytes exhibit substantially less cross- and intra-organ heterogeneity compared to fibroblasts, but nevertheless possess

organotypic features, as reflected by molecular fingerprints and anatomical localization.

**Comparison to other organs**. The qualitative nature of scRNA-seq offers possibilities to compare data collected by different investigators and at different time points. To investigate if the

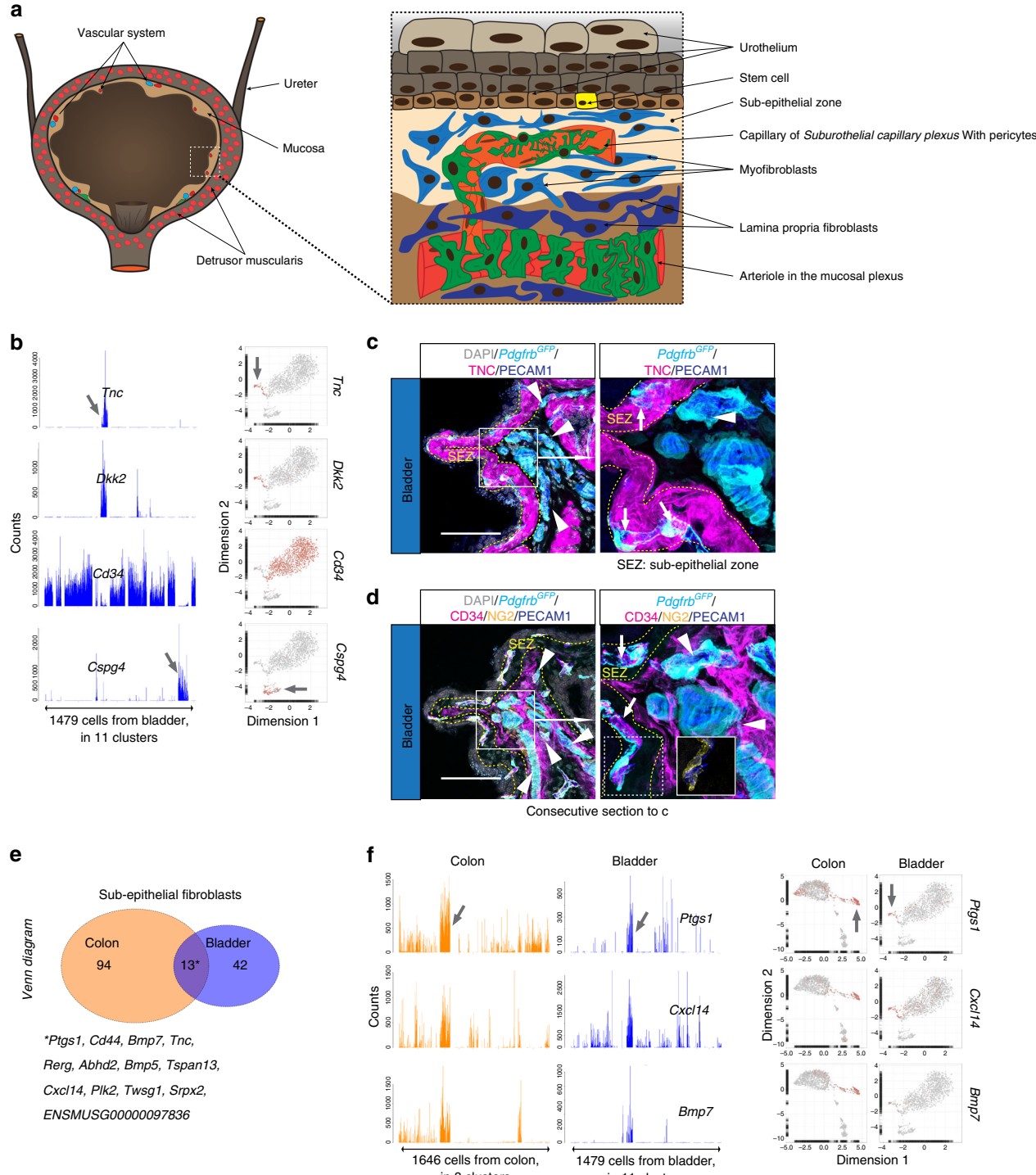

**Fig. 6 Fibroblast subtypes in the bladder. a** Schematic depiction of bladder anatomy. **b** Bar plots and UMAP visualization (gray, low; red, high expression) showing examples of genes with cell subtype-specific expression (arrows). **c**, **d** Immunofluorescence staining of bladder samples from *Pdgfrb^GFP* reporter line for **c** TNC, or **d** CD34, NG2, and PECAM1. Arrows: capillaries of the subepithelial zone, surrounded by TNC+ fibroblasts, arrowheads: large perpendicular vessels in deeper mucosa, surrounded by CD34+ fibroblasts. Scale bar: 100 μm. Consecutive sections were used for **c** and **d**. **e** Venn diagram of *Tnc^+ Cd34^−* (subepithelial fibroblasts) enriched genes from colon and bladder. **f** Examples of commonly subepithelial fibroblast enriched genes presented as bar plots and UMAP (gray, low; red, high expression). Asterisk: gene list of commonly enriched genes (see also Supplementary Table 3).

general patterns of mesenchymal heterogeneity extend also to other organs, we integrated the data with previously published mural cell and fibroblast datasets from brain and lung[11]. In this combined analysis of six organs, the fibroblasts formed distinct

organ-specific clusters, whereas the mural cell clusters were more homogenous and comprised cells from multiple organs (Fig. 8a, Supplementary Data 1). Importantly, many SMC from brain and lung clustered together with the SMC from the other organs,

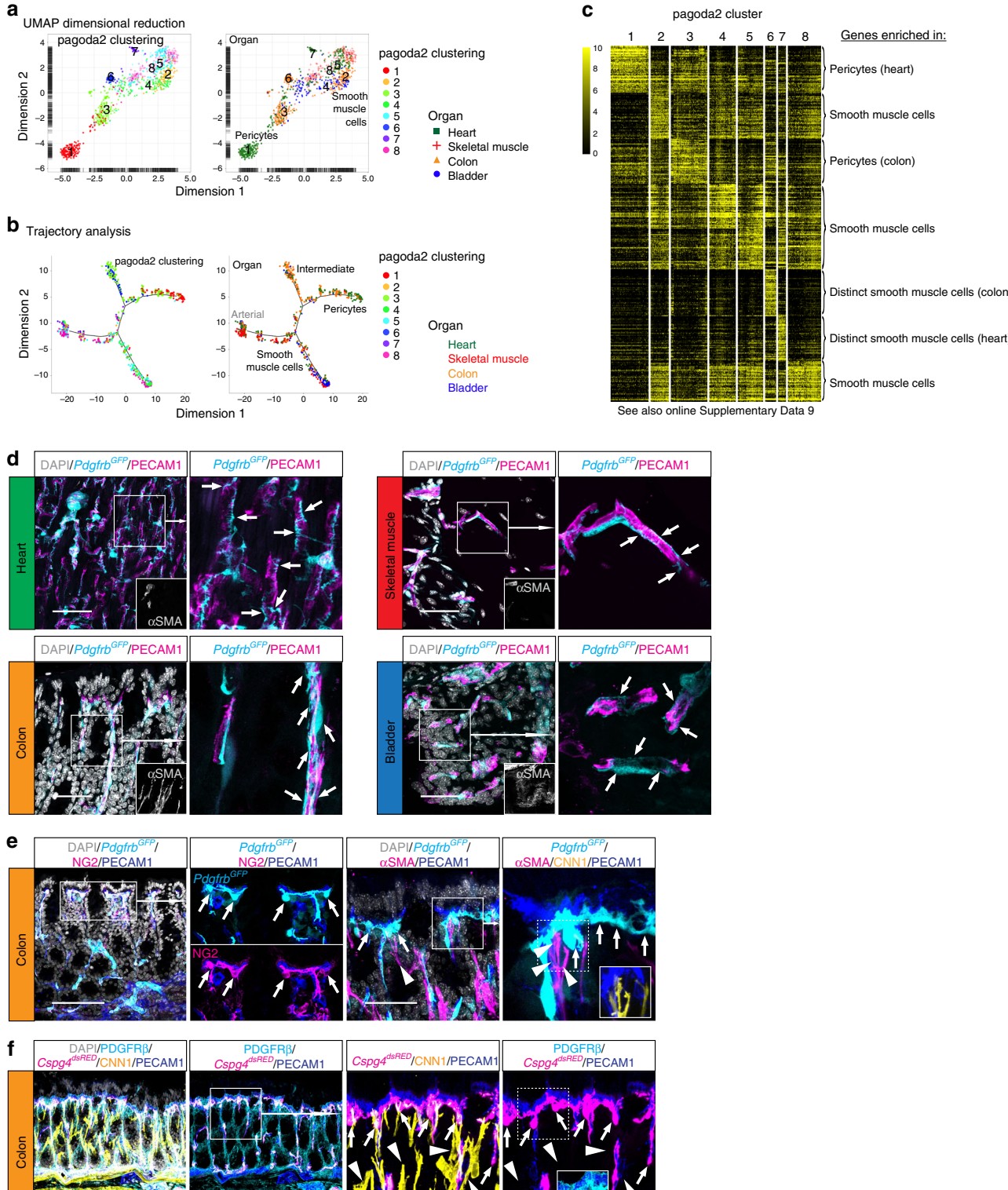

**Fig. 7 Mural cell organotypicity. a** UMAP visualization of the mural cell dataset, color coded and annotated for pagoda2 clustering (left) or color coded for organ of origin and annotated for pagoda2 clustering (right). **b** Trajectory analysis plots of the mural dataset color coded for pagoda2 clusters (left) or organ of origin (right). **c** Expression heat map (black, low; yellow, high) of cluster enriched genes (see also Online Supplementary Data 9). **d** Samples from all four organs from the *Pdgfrb*^GFP reporter line, stained with immunofluorescence for αSMA and PECAM1. Arrows highlight pericytes and their morphological distribution at the organ-specific capillaries. Scale bars: 50 μm. **e** Immunofluorescence staining of colon samples from *Pdgfrb*^GFP reporter line for NG2 (*Cspg4*) and PECAM1 (left panel), or αSMA, CNN1 and PECAM1 (right panel). **f** Immunofluorescence staining of colon samples from *Cspg4*^dsRED reporter line for PDGFRβ, CNN1 and PECAM1. Arrows highlight colon pericytes located at the subepithelial capillary loop, arrowheads indicate αSMA⁺ CNN1⁺, but *Cspg4* (NG2)⁻ interstitial SMC. Scale bars: **e** 100 μm (left panel **f**), 50 μm (**e** right panel).

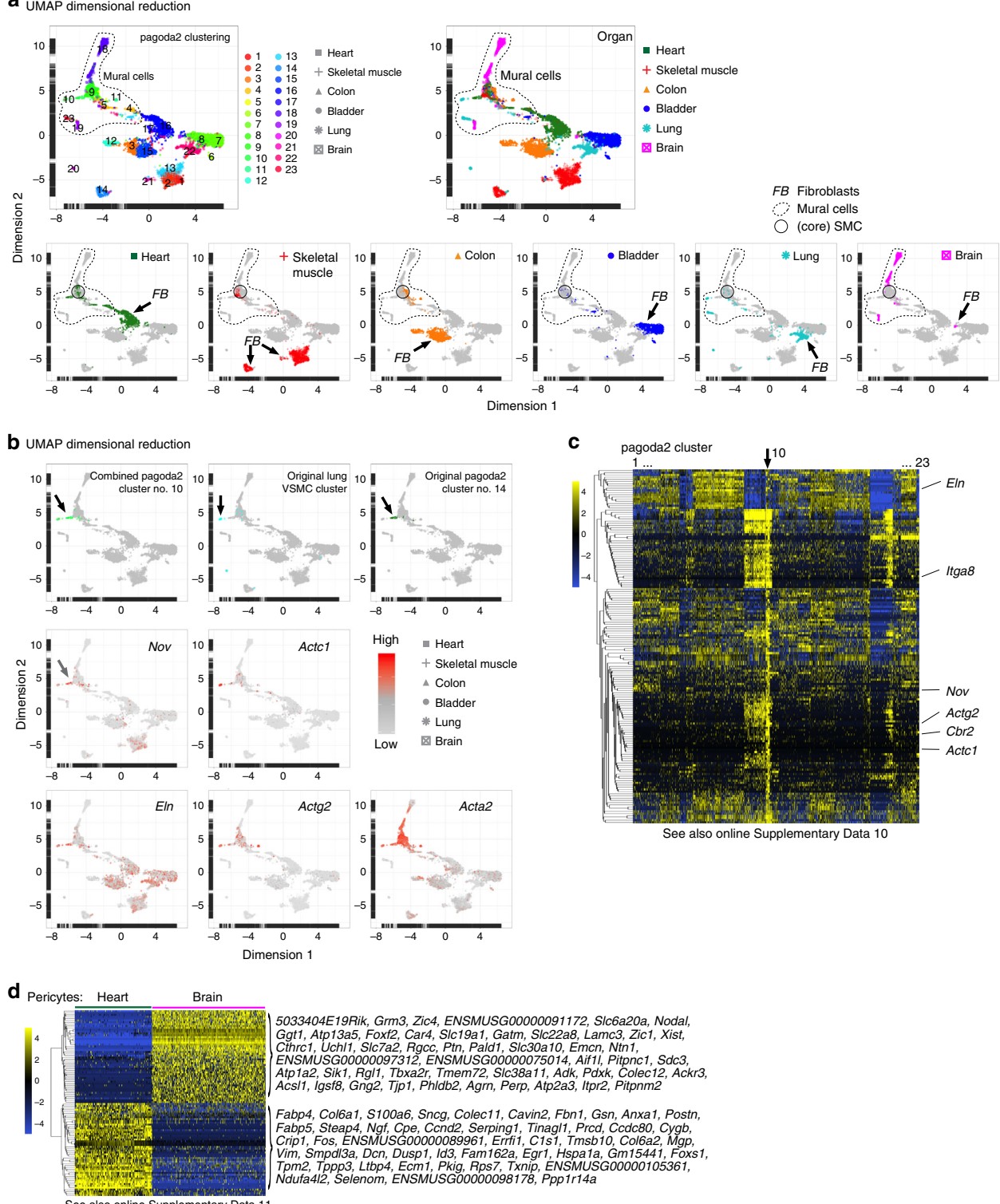

**Fig. 8 Heterogeneity analysis in additional organs. a** UMAP visualization of the combined analysis of the heart, skeletal muscle, colon, and bladder together with the mesenchymal cells of the brain and lung datasets[11], color coded for pagoda2 clusters (upper left) or the organ of origin (upper right). Cell clouds containing mural cells and core SMC are indicated (lower panel). **b** UMAP visualization highlighting cells of pagoda2 cluster 10 (upper left), cells of the lung VSMC cluster (upper middle), cells of the original complete dataset cluster 14 (upper right), or expression levels of cluster 10 of the combined analysis enriched genes (*Nov, Actc1, Eln,* and *Actg2*) as well as *Acta2* (gray, low; red, high). **c** Expression heat map (blue, low; yellow, high) of enriched genes in cluster 10 of the combined analysis (see also Online Supplementary Data 10). **d** Expression heat map (blue, low; yellow, high) of each 50 most differentially expressed genes in heart vs. brain pericytes (see also Online Supplementary Data 11).

suggesting a high degree of cross-organ vascular SMC similarity (Fig. 8a). A small but distinct population of SMC obtained from heart (the $Sost^+$ and $Cbr2^+$ cells described in Fig. 7 and Supplementary Fig. 9c) co-clustered with vascular SMC from the lung[11] (Fig. 8b). These SMC showed enriched expression of a number of genes, including cellular communication factor 3 (*Nov*/*Ccn3*), cardiac muscle actin α-1 (*Actc1*), elastin (*Eln*), smooth muscle actin γ-2 (*Actg2*), and integrin alpha-8 (*Itga8*) (Fig. 8b, c, Supplementary Data 10) and likely represent SMC from the heart's large vessel outlets, i.e., the pulmonary trunk and aorta[49,50]. Comparison of the two most abundant pericyte populations from heart and brain revealed a substantial organotypicity (Fig. 8d, Supplementary Data 11), which is in agreement with our previous comparison of brain and lung pericytes[11,51].

Using other datasets, we tested the predictive capacity of the 90-gene signature (Fig. 1f) to distinguish fibroblasts and mural cells across organs. This signature confirmed the published discrimination between fibroblasts and mural cells in brain and lung[11] (Supplementary Fig. 11a, Supplementary Datas 12 and 13). We next applied the 90-gene signature to data extracted from Tabula Muris[52] and again found distinct prediction of fibroblast and mural cells (Supplementary Fig. 11b, c, Supplementary Datas 14–21). The 90-gene signature thus seems capable of distinguishing fibroblasts from mural cells across multiple organs.

## Discussion

Despite being first identified more than 150 years ago, fibroblasts have remained nebulous cells with regard to molecular composition, extent of heterogeneity across organs and relationships to other mesenchymal cell types[4]. Here, we characterize fibroblasts from four different muscular organs at single-cell resolution and compare them to vascular mural cells (summarized in Fig. 9). Our data reveal common markers defining fibroblasts in all organs, but also an extensive fibroblast organotypicity, as well as inter-organ heterogeneities. While no singular marker could discriminate all fibroblasts from all mural cells, we provide a 90-gene signature capable of doing so across all organs studied herein, as well as in other public datasets[52]. This signature could therefore be used to resolve ambiguous cell type annotation in yet other single-cell datasets, as well as in tissue analysis using antibody and antisense probes.

Fibroblasts are important producers of ECM at many locations in the body, but how ECM diversity in different organs is generated has been unclear. Our transcriptomic data reveal extensive inter-organ transcriptional heterogeneity among fibroblasts, differences that are particularly obvious regarding expression of genes in the matrisome[23]. The differences in matrisome gene expression were larger than for other gene/protein categories, indicating that fibroblasts specifically tailor the ECM in accordance with the organ-specific physiology[53]. The observation that transcription factors were the least variable in this analysis may suggest that epigenetic differences contribute to fibroblast heterogeneity, something that will require further investigation.

Concerning the classification of cell type, subtype, and/or state based on single-cell transcriptomics, it should be remembered that the numbers of clusters assigned are to some extent arbitrary and may also reflect the introduction of experimental artefacts, such as cell activation or damage during their isolation. Nevertheless, the combination of cellular dispersion in UMAP and in situ mapping of anatomical localization of cells using relevant markers, as done herein, should provide physiologically meaningful annotations. For example, eight bladder fibroblast clusters formed one inter-dispersed cloud in the UMAP, leading us to conclude that these eight clusters correspond to one major fibroblast type, however, with small molecular differences that

caused the multiple cluster assignment. On the other hand, we also found clear signs of intra-cluster zonation, which often is concealed by two-dimensional nearest neighbor clustering methods, and thus requires an additional analysis, e.g., by SPIN, to become recognizable. We find this in the colon fibroblast populations that distribute along the crypt–surface axis, while there are clear differences between the extreme states located at the apex (mucosal surface) and base (mucosal bottom) of the crypts, gradual changes, and intermediate phenotypes (zonation) occur at middle locations along the crypt–surface axis.

In contrast to the fibroblasts, vascular mural cells showed considerably less cross-organ heterogeneity with the SMC exhibiting the least heterogeneity, however, with the exception of two specialized SMC populations—the colonic interstitial SMC and the SMC of the large vessel trunks entering and exiting the heart. Pericytes were more heterogeneous than SMC and displayed distinct signs of organotypicity, corroborating previous observations from brain and lung[11]. Pericytes in different organs differed in terms of expression of transporters, or of components of the SMC contractile machinery such as *Myh11*, *Tagln*, and *Acta2*. Heart pericytes displayed low or undetectable levels of contractile components similar to what we have previously reported for brain and lung[11]. Pericytes from colon, and especially the bladder[54], however, expressed several SMC contractile markers. Pericytes also exhibited interesting morphological differences across organs. At subepithelial capillary loops in the colon they displayed a stereotypic positioning on the far side of the capillaries in relation to the epithelial lumen, possibly suggesting that local morphogenic cues instruct pericyte differentiation as well as positioning.

The characterization of distinct subtypes of fibroblasts and mural cells revealed intra-organ heterogeneity as well as cross-organ similarities. In the skeletal muscle, in addition to the previously described perimysial cells[24,27], we identified an additional cell type of fibroblast: the paramysial cells located at the fringe of the perimysium fasciae. The peri- and paramysial cells differ in terms of expression of ECM components. The perimysial cells appear molecularly very closely related to the $Scx^+$ $Tnmd^+$ cell population identified in a recent study[55], which however, were suggested to reside in the skeletal muscle parenchyma. We map these cells primarily to muscular fasciae. Tenocytes recently characterized in the patella tendon[56] exhibit a similar molecular fingerprint as perimysial cells in agreement with the notion that the perimysium is continuous with tendons[26].

Cross-organ resemblance is exemplified by the molecular similarities between the perimysial cells and cardiac valve interstitial cells[31,32], including shared expression of cartilage- and tendon-associated genes such as *Wif1*, *Comp*, and *Fmod*. Therefore, it may be speculated that cardiac valve calcification[32,57] and heterotopic ossification of the skeletal muscle[58] have common molecular determinants. Another cross-organ similarity regarded a small population of SMCs isolated from heart (this study) and lung[11]. This may suggest a common anatomical origin (the cardiopulmonary vessel trunks).

The classification of fibroblast subtypes also unraveled common cell zonation principles between the colon and the bladder. In the colon, $Tnc^+$ $Cd34^-$ fibroblasts reside close to the surface at the crypt apex, while $Tnc^-$ $Cd34^+$ cells are located at deeper mucosal areas (crypt base). A similar zonation was revealed in the bladder, where $Tnc^+$ $Cd34^-$ cells are present in the sub-urothelial mesenchyme, while $Tnc^-$ $Cd34^+$ cells locate deeper down in the bladder mucosa. This organizational similarity between colon and bladder extends also to genes encoding components of the BMP and WNT signaling pathways[37,59,60]. The fibroblasts residing at the crypt apex of the colon are similar to a previously described stromal cell population with low responsiveness to chemically

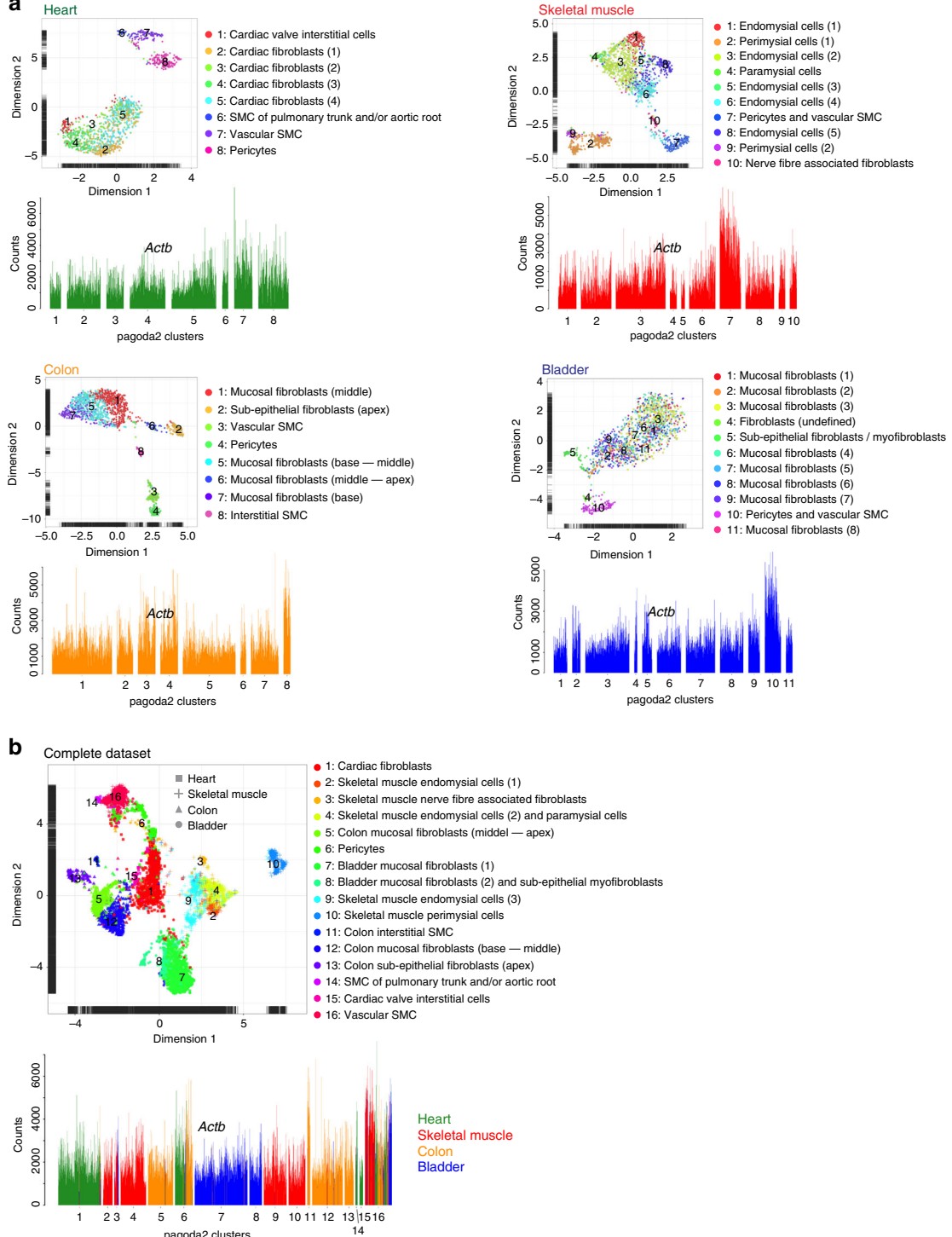

**Fig. 9 Cell annotation summary. a** UMAP visualization with pagoda2 cluster annotation and bar plots showing *Actb* expression of organ specific datasets. **b** UMAP visualization with pagoda2 cluster annotation and bar plot showing *Actb* expression with organ of origin color coded in the complete dataset.

induced colitis (Str2 cells)[41]. We find that transcription factor *Foxl1*, which is important for intestinal development and maintenance[42,61], is expressed by these cells. This is in agreement with a recent report[41], but contrasts with another study that reported *Foxl1* expression around the stem cell niche at the crypt base in the small intestine[62], discrepancies that might reflect differences between the small and large intestine. The epithelial zonation along the crypt–surface axis is well appreciated and it is intriguing

to speculate about the crosstalk and reciprocal signaling mechanisms between the zonated epithelial and mesenchymal cell pools. PDGFRα-positive mesenchymal cells recapitulating crypt apex cells were shown to exhibit an important function during villi expansion[34], and it is likely that the different fibroblast populations along the crypt–surface axis are involved in adult intestinal homeostasis, stem cell niche function, and response to injury or damage[43,63]. Extended analysis combining and utilizing

published datasets of intestinal epithelial cells and mesenchymal cells with regard to ligand–receptor pairs could shed further light on the intimate signaling mechanisms that operate at the epithelial–mesenchymal interface of the intestine and the shaping of the cellular zonation gradients.

Dysregulation of fibroblasts has been implicated in several diseases, including fibrosis, cancer, inflammation, and cardio-vascular diseases[64]. Along these lines, mutations in type-XII collagen α-1 (*COL12A1*) were recently identified to cause hypo-tonia, muscle weakness, and joint hypermobility[65,66]. Our finding that *Col12a1* is highly expressed in perimysial and paramysial cells thus implicates a role for these cells in the structural integrity and function of skeletal muscle. Furthermore, *Col12a1* is expressed by fibroblasts of the colon and bladder lamina propria, suggesting that *COL12A1* mutations may cause pathological changes of the ECM also in these organs. Moreover, an activated fibroblast population exhibiting a molecular fingerprint similar to the perimysial cells, including *Wif1* expression, was recently found at the infarct border zone after myocardial infarction[67]. In conclusion, the data presented in this study, which are available as a transcriptional repository at [https://betsholtzlab.org/Publications/FibroblastMural/database.html], provide molecular identities to important, yet poorly understood and annotated mesenchymal cell types. This information will be useful to better understand the role of these cell types in health and in a variety of pathological processes.

## Methods

**Animals.** All animal experiments were carried out in accordance with Swedish legislation and local guidelines and regulations for animal welfare and were approved by Linköpings Animal Research Ethics committee [Linköpings djurför-söksetiska nämnd], approval ID 729. All animals were housed in standard, single ventilated cages with 12 h light–12 h dark cycle, and had ad libitum access to water and chow. The house temperature was kept at $20 \pm 2\,°C$ and the relative humidity was kept as $50 \pm 5\%$. In this study we used the following mouse strains; *C57Bl6* (The Jackson Laboratory, *C57Bl6/J*, maintained as breeding colony at the local animal facility), *Pdgfrb^GFP* (Genesat.org, Tg(*Pdgfrb*-eGFP) JN169Gsat/Mmucd), *Pdgfra^H2BGFP* ((*Pdgfra^tm11(EGFP)Sor*), a gift from P. Soriano), *Cspg4^dsRED* (The Jackson Laboratory, Tg(*Cspg4*-DsRed.T1)1Akik/J), *Acta2^GFP* (The Jackson Laboratory, Tg(*Acta2*-GFP)1Pfk), *Cldn5^GFP* (Tg(*Cldn5*-GFP)Cbet/U), and combi-nations of these strains. All mice were backcrossed to the *C57Bl6/J* background. For the experiments, adult male mice at an age of 10–20 weeks were used.

**Isolation of single cells from murine tissues.** The protocol below was used for all tissues. Mice were euthanized by cervical dislocation and the organ(s) (heart, skeletal muscle, bladder, colon) of interest were dissected out and immediately placed into ambient phosphate-buffered saline (PBS) solution (DPBS, Thermo-Fisher Scientific). In case of the skeletal muscle; gastrocnemius including soleus, or both muscles separated were harvested for further processing. In one instance the soleus muscles from two mice (one *Pdgfra^H2BGFP* and one *Pdgfrb^GFP*) were pooled prior to dissociation for single cell suspension preparation. The tissues were then cut into smaller pieces and incubated in dissociation buffer (Skeletal Muscle Dis-sociation kit from Miltenyi, supplemented with 1 mg/ml Collagenase type IV from Sigma) at 37 °C with horizontal shaking at 500–800 rpm. For isolation of *Acta2* positive cells from *Acta2^GFP* reporter mice, elastase (~2.5 U/ml, Sigma) was added to the dissociation buffer. Three to four cycles of physical disintegration by pipetting were applied with 10 min intervals to the samples. Thereafter the cell suspension was sequentially passed through a 70 μm and a 40 μm cell strainer. The 70 μm cell strainer was additionally washed with 5 ml of DMEM (ThermoFisher Scientific). Cells were pelleted by centrifugation at $300 \times g$ for 5 min, supernatant was removed and the cell pellet resuspended in FACS buffer (PBS supplemented with 0.5% bovine serum albumin, 2 mM EDTA, 25 mM HEPES). For antibody labeling, the cell suspension was incubated with the respective, fluorophore-conjugated antibody (anti-PDGFRα, anti-PDGFRβ, and anti-CD31) for 20 min at RT, then centrifuged for 3 min at $240 \times g$, supernatant removed, and the cell pellet was resuspended in FACS buffer and placed on ice.

**Fluorescent activated cell sorting (FACS).** Antibody-stained cell suspensions were analyzed using Becton Dickinson FACSAria III or FACSMelody cell sorters equipped with a 100 μm nozzle, and single cells meeting the selection criteria as described below were deposited into 384-well plates containing 2.3 μl lysis buffer (0.2% Triton X-100, 2 U/ml RNase inhibitor, 2 mM dNTPs, 1 μM Smart-dT30VN, ERCC $1:4 \times 10^4$ dilution). Of note, the analysis and cell sorting by FACS was not

used for cell type identification, but for enrichment and capture of cells expressing gene or protein signatures of interest; *Pdgfra* (PDGFRα)⁺ or *Pdgfrb* (PDGFRβ)⁺, and *Pecam1* (CD31)⁻. For single cell sorting; first, a gate for forward scatter-area/side scatter-area (FCS-A/SSC-A, linear scale) was set generously around present events only excluding events with low values (cell debris and red blood cells). Second, doublet discrimination was implemented using FCS-A/FSC-height and SSC-A/SCC-height, however, with a generous threshold for the distance of events from the diagonal line, to prevent the introduction of a bias toward round shaped cells. Third, selected events were then analyzed for positive fluorescent signals, while 'fluorescence minus one' antibody-stained samples, or cells prepared from mice lacking fluorescent reporters, were used to ensure correct gating and as negative controls. To enrich for mesenchymal cell populations, either staining with antibodies anti-PDGFRα or anti-PDGFRβ, or transgenic reporter mouse strains *Pdgfra^H2BGFP*, *Pdgfrb^GFP*, *Acta2^GFP*, *Cspg4^dsRED*, or combinations of the mouse strains were applied, together with antibody staining for CD31, to avoid cell-doublet selection. Cells positive for either of the aforementioned mesenchymal cell markers that were also positive for CD31 were excluded from the sort. Plates were briefly centrifuged prior to sorting, while correct deposition of the droplet into the 384-well plate (aiming) was controlled by test-spotting of beads onto the seal of the respective plate, and if necessary the plate position was adjusted for each new plate placed into the machine. The sample and plate holder of the FACS machine were maintained at 4 °C, and the plates were placed on dry-ice immediately after the sorting was completed and subsequently stored at −80 °C until downstream processing.

**SmartSeq2 library preparation and sequencing.** Single-cell cDNA libraries were prepared according to the previously described Smart Seq2 protocol[16]. In brief, mRNA was transcribed into cDNA using oligo(dT) primer and SuperScript II reverse transcriptase (ThermoFisher Scientific). Second strand cDNA was syn-thetized using a template switching oligo. The synthetized cDNA was then amplified by polymerase chain reaction (PCR) for 23–26 cycles, depending on the tissue-origin of the respective mRNA sample. Purified cDNA was quality con-trolled (QC) by analyzing on a TapeStation 4200 or 2100 Bioanalyzer with a DNA High Sensitivity chip (Agilent Biotechnologies). When the sample passed the QC, the cDNA was fragmented and tagged (tagmented) using Tn5 transposase, and each single well was uniquely indexed using the Illumina Nextera XT index kits (Set A–D). Thereafter, the uniquely indexed cDNA libraries from one 384-well plate were pooled into one sample to be sequenced on one lane of a HiSeq3000 sequencer (Illumina), using dual indexing and single 50 base-pair reads.

**Sequence data processing.** Pooled single-cell cDNA library samples were sequenced as described above. Demultiplexing into single-cell fastq files was per-formed applying standard parameters of the Illumina pipeline (*bcl2fastq*) using Nextera index adapters. The individual fastq files were then mapped to the mouse reference genome GRCm38 (mm10), using TopHat2 with Bowtie1 or Bowtie2 option[68,69], where adapter sequences were removed using trim galore before read mapping. Doublets were removed using the samtools software. The generated BAM files containing the alignment results were sorted according to the mapping position, and raw read counts for each gene were calculated using featureCounts from the Subread package[70]. For technical control, 92 ERCC RNAs were included in the lysis buffer and in the mapping.

Thereafter, the cells were combined in the expression matrix showing raw counts per gene for each individual cell as input data. Annotation of the ENSEMBLE identifiers was done using the org.mM.eg.db package (version 3.7.0) in R-software, keeping the ERCC counts to be used as technical controls in the dataset.

To calculate general attributes of the expression matrix the SingleCellExperiment R-software package was applied[71]. Filtering of low quality cells was done in a stepwise manner. First cells with low library sizes (≤50,000 counts), and low number of expressed genes (≤1500) were removed from the dataset. If necessary, also cells with high percentage of reads mapped to ERCCs or mitochondrial genes (both > 10%) were removed. Cells with a high number of expressed genes (≥10,000) were removed as potential doublets. Non-, or low-expressed genes were removed; genes had to fulfill the qualification criteria: gene expression in at least three cells with a counts value >20, and a cumulative counts value of 300 to be retained in the dataset. Unintentionally collected cells that did not express either *Pdgfra* or *Pdgfrb* were then removed from the dataset. Cells that displayed a clearly contaminated transcriptome with, e.g., endothelial cell or immune cell-specific gene signatures were also removed from the dataset. This resulted in a final dataset for analysis, composed of cells from $n = 24$ different male mice (Supplementary Fig. 1c, Supplementary Table 4). After sequencing and quality control a dataset of 6158 single-cell transcriptomes, comprised of 1279 cells from the heart, 1754 cells from the skeletal muscle, 1646 cells from the colon, and 1479 cells from the bladder was constructed and used for bioinformatics analysis.

The dataset was organized in the SingleCellExperiment R-software package[71]. The pathway and gene set overdispersion analysis (*pagoda2*) R-software package (https://github.com/hms-dbmi/pagoda2)[17] was applied to perform principle component analysis (PCA; using the attributes nPca = 100, n.odgenes = 3000) and nearest neighbor clustering. For dimensional reduction visualization the

UMAP function was applied (UMAP: uniform manifold approximation and projection)[20].

For construction of the bar plot database, the cluster information from pagoda2 was used. For in-cluster cell distribution the SPIN algorithm[18,19] was applied using the 1000 most differentially expressed genes per cluster (/backspin –i input.cef –o output.cef –f 1000 –b both). The counts values were normalized to 500,000 counts library size per cell for visualization in bar plots.

For the generation of the organ-specific datasets the same parameters, as described above, were used for the calculation of PCA and UMAP, as well as pagoda2 clustering and SPIN in-cluster cell distribution.

Differential gene expression analysis was performed using the monocle R-software package[72]. Pairwise comparison between selected groups of cells was done using the differentialGeneTest function and the basic selection criteria: 100 counts per cell as threshold for gene detection, expression in ≥30% of cells per group. For the differential expression analysis between fibroblasts and mural cells (Fig. 1e, f), as well as for the identification of specific fibroblast subpopulation per organ (Fig. 4e and Fig. 6e), in addition, a greater than equal to twofold ($log_2$) difference in expression was used for gene selection. For the molecular diversity analysis of perimysial cells (Fig. 3f, Supplementary data 7), the same criteria as above were applied, except a greater than equal to onefold ($log_2$) difference in expression was used for gene selection. For the genes presented in Supplementary Fig. 3b and Fig. 7c each cluster was compared with the rest of the dataset and the 50 most differentially expressed genes were selected for each cluster, and genes (unique) are plotted in the heat map. The cross-referencing of differentially expressed genes in selected cell populations was visualized using venn diagrams, drawn using the VennDiagram R-software package (https://CRAN.R-project.org/package=VennDiagram).

For gene-category restricted UMAP analysis, we obtained gene lists of GO—terms from the MGI server (http://www.informatics.jax.org), or published work[11,22,23]. Used GO-terms: cytoskeleton: GO_0005856, cell activation: GO:0001775, cellular response to cytokine stimulus: GO:0071345, cell surface receptor signaling pathway: GO:0007166, cell–cell signaling: GO:0007267, ECM: GO:0031012. These gene lists were used to sub-set the SingeCellExperiment object to retain only expressed genes from the GO gene lists, creating a new SingleCellExperiment object. Thereafter, PCA was recalculated using the same attributes as before and UMAP visualization performed using meta data of cellular origin or pagoda2 cluster affiliation.

WNT (GO:0016055) and BMP (GO:0030509) signaling pathway gene lists were collected from the MGI server (in total 553 genes). For zonation analysis of WNT and BMP signaling pathway genes in the colon, the fibroblast clusters (# 1, 2, 5–7) were extracted and a SPIN range calculated using genes with a cumulative expression >100 counts. For the fibroblast zonation analysis of the colon (Supplementary Fig. 7b, Supplementary Data 8), the applied thresholds when using the differentialGeneTest function were set to 30 counts per cell for gene detection and ≥25% of cells per group, due to the overall lower expression of genes belonging to the BMP and WNT signaling pathways. Further, WNT and BMP signaling pathway genes with an accumulative count value ≥1000 were kept in the colon fibroblast dataset (66 genes) for visualization. The gene expression count values were then fitted to a smooth curve for each gene using the loess function with default parameters in R-software and visualized in a heat map.

**Immunofluorescence staining**. Standard methods for immunostaining were applied. In brief, tissues were harvested from euthanized mice as described before and fixated by immersion in 4% formaldehyde for 4–12 h at 4 °C, followed by immersion in 20% sucrose/PBS solution for at least 24 h at 4 °C. Thereafter, tissues were embedded for cryo-sectioning and sectioned on a CryoStat NX70 (Thermo-Fisher Scientific) to 14 or 30 μm thick sections collected on SuperFrost Plus glass slides (Metzler Gläser). Sections were stored at −80 °C. For staining, sections were allowed to thaw at RT. After thawing, sections were blocked for >60 min at RT with blocking-buffer (serum-free protein blocking solution, DAKO), supplemented with 0.2% Triton X-100 (Sigma Aldrich). Thereafter, sections were sequentially incubated with primary antibodies and corresponding fluorescently conjugated secondary antibodies (Supplementary Table 5) according to manufacturer's recommendations. Sections were mounted with ProLong®Gold mounting medium, containing 4,6-diamidino-2-phenylindole (DAPI, ThermoFisher Scientific), or ProLong®Gold without DAPI, when Hoechst 33342 (trihydrochloride, trihydrate, ThermoFisher Scientific) was applied at 10 μg/ml together with the secondary antibodies. Micrographs were acquired using a Leica TCS SP8 confocal microscope (Leica Microsystems). Images were graphically handled, and adjusted for brightness and contrast using ImageJ/Fiji software[73]. If not otherwise stated, maximum intensity projections of acquired z-stacks are shown in the figures.

**RNA in situ hybridization (RNAscope®)**. For in situ hybridization, the RNA-scope® system (Advanced Cell Technologies) was applied according to the manufacturer's protocol. In brief, tissue sections were obtained as described before (Immunofluorescence staining). After dehydration, the sections were incubated with Pretreat 4 solution for 30 min at RT. Then, RNAscope® probes (Supplementary Table 5) were hybridized on the sections for 2 h at 40 °C, and thereafter the fluorescent detection protocol (Amplification-FL) was applied according to the manufacturer's recommendations. Sections were mounted with ProLong®Gold

mounting medium. Micrographs were acquired and processed for visualization as described above (see Immunofluorescent staining).

**Capillary diameter quantification**. For quantitative determination of capillary diameter, maximum intensity projections of micrographs (see Immuno-fluorescence staining) from anti-PECAM1 stained tissue sections were used. For the colon, the subepithelial capillary loop was excluded of the measurement, due to its specialized localization/properties. The capillary diameter was measured using the line tool in ImageJ/FIJI software. First, the average diameter per micrograph was calculated. Second, the average diameter per individual mouse was calculated, which then is used as one biological replicate ($n$) for $p$ value calculation. One-way ANOVO with Tukey's test for multiple comparisons was used to determine statistical significance.

**Statistics and reproducibility**. In this section, it is stated how often experiments have been repeated independently and obtained similar results. All antibody immunofluorescence experiments were performed at least two times using identical or varying antibody combinations, in total analyzing tissue samples from at least three individual mice. All RNAscope experiments were performed at least two times, in total analyzing tissue samples from at least three individual mice. Statistical analysis of capillary diameter (Supplementary Fig. 10e) quantification was performed with GraphPad Prism 8.

**Reporting summary**. Further information on research design is available in the Nature Research Reporting Summary linked to this article.

## Data availability

All data to support the findings of this study are included in the paper, the Supplementary Information and freely available as a searchable database at (https://betsholtzlab.org/Publications/FibroblastMural/database.html). Further data are available from the corresponding authors upon reasonable request. The single-cell RNA-sequencing raw data of this study have been deposited in the NCBI's Gene Expression Omnibus database under the accession number: GSE150294.

The source data underlying Supplementary Fig. 10e are provided in the Source Data file.

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

## Acknowledgements

We thank Drs. Xidan Li for bioinformatics support, and Rahgu Kalluri for valuable discussions and suggestions, the P.K.L. Huddinge, the MedH FACS facility, and Tove Berg, Bora Baskaner, and Åsa Hedberg for technical help and support. This study was supported by grants from the Swedish Science Council, the Swedish Cancer Foundation, the Knut and Alice Wallenberg Foundation, The Leducq Foundation, The Louis-Jeantet

Foundation, The National Nature Science Foundation of China (81870978) and Astra-Zeneca AB. Open access funding provided by Karolinska Institute.

## Author contributions

L.M., G.G., U.L., and C.B. conceived the study and designed the project and experiments. L.M., G.G., S.L., J.L., and G.M. performed the experiments. S.L., G.M., and L.M. designed and performed FACS experiments. J.L.M.B. and E.M.H. supervised the work performed by G.M. and S.L., respectively. J.L., S.G., B.B., I.V.C, Å.S., and E.R. performed library preparation for single-cell RNA-sequencing and J.L. performed the sequencing. L.M., G.M., L.H., and Y.S. performed bioinformatics analysis. L.H. and Y.S. constructed the online database. L.M., G.G., M.V., U.L., and C.B. analyzed the bioinformatics data. L.M., U.L., and C.B. wrote the paper with critical input from X.-R.P., and L.M. created the figures with help from S.L., L.H., and Y.S. All authors reviewed the paper.

## Competing interests

X.-R.P. is an employee of AstraZeneca, C.B. is a consultant for AstraZeneca. The remaining authors declare no competing interests.
