## [Peer Review File · Nature Communications]

Reviewers' comments:

Reviewer #2 (Remarks to the Author):

The authors need to be applauded for substantially improving the manuscript by adding an array of new analyses that were necessary to demarcate poorly described mesenchymal cell types and to provide a more detailed inter-organ analysis (particularly regarding fibroblasts). They have now successfully defined a panel of marker genes that can be used to infer mesenchymal cell identity in single cell datasets. The authors further report fibroblasts to be extremely heterogeneous with the matrixome being the driving factor, but also show inter-organ similarities (heart valves and skeletal muscle). Interestingly, fibroblasts seem to be zoned in multiple tissues, which is in line with previous findings in other cell types (epithelial cells, hepatocytes, endothelial cells). Pericytes, in contrast, seem to be more homogenous and do not show major inter-organ heterogeneity. The manuscript includes several high-quality images validating *in silico* findings. The manuscript is considered highly suitable for the broad readership of the journal. In further advancing their work, the authors are encouraged to consider the following minor comments:

1. The authors report mural cells to be more homogenous than fibroblasts. Although this finding seems very interesting at first, it cannot be excluded that this may rather be explained by artefacts introduced by employing low-depth scRNA-Seq. As clustering approaches take the variable gene expression across all genes into account, the degree to which cellular heterogeneity can be assessed depends on total genes and gene counts identified on a per cell basis. If the median nGene/cell is lower for mural cells than for fibroblasts, the pool of variable genes that can be used for clustering is restricted, which could lead to underappreciation of heterogeneity. This phenomenon can be observed with other low RNA containing cells (e.g. endothelial cells, for which a pronounced core transcriptome seems to mask variable gene expression), where clustering approaches fail to capture heterogeneity effectively (especially if gene expression changes gradually/is zoned). One way to circumvent this problem could be iterative and random downsampling of genes taken into account for clustering to equal numbers for mural cells and fibroblasts.

2. Fig. 3f: The authors show that gene expression is zoned for perisymal cells. It is not clear, however, alongside which axis and in which direction (as done in Fig. S7b))

3. Fig. 4 displays fibroblast heterogeneity in the heart and cross-references a *Wif1* expressing subpopulation (that shows strong similarity with valve interstitial cell types that have previously been published) with transcriptionally similar cells from the skeletal muscle (perimysial cells). While this clearly illustrates the power of multi-organ high-resolution atlases, it is a little strange that the other subpopulations are not at all characterised or even mentioned in the text.

4. The authors elegantly show that fibroblast gene expression is zoned in a similar manner as for epithelial cells in the colonic mucosa. While this is a very interesting finding, it also makes room for new questions, e.g. is the zonation complementary or additive between cell types?, which cell type is instructive for establishing zonation?, what are the key molecules for establishing zonation? In-depth analyses might be beyond the scope for the manuscript, but the authors could still hypothesize and discuss a bit more how they envision the zonation to be established.

5. The authors obtained only a very low number of pericytes from skeletal muscle or bladder and argue that this might be due to inefficient tissue dissociation. This explanation, as it stands, does not seem credible. As SMART-Seq2 is a plate-based scRNA-Seq platform and the authors used FACS sorting to enrich and purify cells, this would mean that the authors were only able to sort 10 cells from the whole organ. Is that correct or did the authors sort a larger number of cells and library generation was only successful for 10 cells? In the latter case, could the authors comment on what they think is the issue generating libraries from muscle/bladder pericytes as compared to other organs (RNA content, sensitivity)?

6. The authors show in a very elegant approach that the matrisome is what drives fibroblast heterogeneity. As total transcription factors expression did not seem to contribute much to the observed heterogeneity, the authors could use a reverse approach by running motif enrichment analyses for the differentially expressed matrix modulators in order to identify key transcription factors that drive heterogeneity (it is clear that the signature of a handful of differentially expressed transcription factors will be “diluted” within the bulk set). Furthermore, it would add to the understanding of pericyte heterogeneity if a similar analysis was performed.

7. Fig. 8: The authors integrate their current work with a previously published dataset on lung and brain fibroblasts/pericytes. Clustering revealed that lung and brain pericytes cluster into separate subpopulations, whereas the organs analyzed in this work rather fall into one cluster. Is this due to batch-effects or does it mirror fundamental differences in biology and function?

8. The authors used 24 individual mice. Fig S1c is supposed to clarify which cells in the cluster were derived from which mouse. Nevertheless, it is impossible to distinguish 24 distinct colors in the UMAP. The authors should include a table depicting how many cells came from which mouse per cluster.

9. It is at times difficult to follow which pagoda cluster annotations were used in the Figures (e.g Fig. S2a, which I assume derives from Fig. 1).

Reviewer #4 (Remarks to the Author):

The authors have addressed my concerns and this work represents a useful resource for the field.

Reviewer #5 (Remarks to the Author):

In general the response to reviewers were positive on many key problems in the original version, probably most importantly dealing with sample size. This certainly provides added rigor to an already very detailed and technically superb manuscript. Thus, the paper is improved.

However, a residual question by two reviewers were not answered. Essentially, with vast number of manuscripts with scRNAseq data, many focused on fibroblasts and mural cells already, how does this manuscript stand apart from the others? Two of the reviewers suggested functional studies or differences in disease states. Fibroblasts are key cell types in many disease states, and as such, their activation is probably their most important criteria. The authors did not add any data to this major critique and instead replied their data set will be important for fibroblast studies in general and differentiating from mural cells. This is true, but without functional data, or the potential changes in a disease state, the work remains descriptive.

Another lingering component that adds to the descriptive “feel” of the manuscript is there is no single marker to identify fibroblasts across organ systems. This is not the fault of the authors for the data, but it is likely something could have been done to analyze this further; e.g., in a particular disease states, does a common marker for fibroblasts appear? Adding more organs to the analysis didn’t necessarily strengthen the initial results, it just added another level of complexity to their story (for better or for worse).

Single-cell analysis uncovers inter- and intra-organ fibroblast heterogeneity and provides criteria for fibroblast identification and discrimination from vascular mural cells.
Muhl et al,

Point-by-point reply:

Reviewer #2 (Remarks to the Author):

The authors need to be applauded for substantially improving the manuscript by adding an array of new analyses that were necessary to demarcate poorly described mesenchymal cell types and to provide a more detailed inter-organ analysis (particularly regarding fibroblasts).

We thank the reviewer for appreciating our work to improve the study.

They have now successfully defined a panel of marker genes that can be used to infer mesenchymal cell identity in single cell datasets. The authors further report fibroblasts to be extremely heterogenous with the matrisome being the driving factor, but also show inter-organ similarities (heart valves and skeletal muscle). Interestingly, fibroblasts seem to be zonated in multiple tissues, which is in line with previous findings in other cell types (epithelial cells, hepatocytes, endothelial cells). Pericytes, in contrast, seem to be more homogenous and do not show major inter-organ heterogeneity. The manuscript includes several high-quality images validating in silico findings. The manuscript is considered highly suitable for the broad readership of the journal.

We again thank the reviewer for the overall positive comments.

In further advancing their work, the authors are encouraged to consider the following minore comments:

1. The authors report mural cells to be more homogenous than fibroblasts. Although this finding seems very interesting at first, it cannot be excluded that this may rather be explained by artefacts introduced by employing low-depth scRNA-Seq. As clustering approaches take the variable gene expression across all genes into account, the degree to which cellular heterogeneity can be assessed depends on total genes and gene counts identified on a per cell basis. If the median nGene/cell is lower for mural cells than for fibroblasts, the pool of variable genes that can be used for clustering is restricted, which could lead to underappreciation of heterogeneity. This phenomenon can be observed with other low RNA containing cells (e.g. endothelial cells, for which a pronounced core transcriptome seems to mask variable gene expression), where clustering approaches fail to capture heterogeneity effectively (especially if gene expression changes gradually/is zonated). One way to circumvent this problem could be iterative and random downsampling of genes taken into account for clustering to equal numbers for mural cells and fibroblasts.

The reviewer raises an important point, which we agree is central to our study. The clustering method used in this study (the pagoda2 pipeline) is indeed based on the most variable genes detected by the algorithm (usually ≈3000 genes). To minimize the introduction of artifacts caused by limited or differential sampling of mRNA from the different cell types, we already previously applied stringent quality filtering criteria (library size > 50000 counts, > 1500 expressed genes, < 10% ERCC, < 10% mitochondrial gene expression) for cell-inclusion, which are now better explained in the manuscript (page 30, line 928-933). In the included cells, the median number of expressed genes per cell are 3970 for mural cells and 4730 for fibroblasts, which is proportional to the total (cumulative) number of genes expressed (14078 in mural cells (clusters 6,11,14,16) and 17908 in fibroblasts (all other clusters)). Hence, we detect more genes/cell in fibroblasts likely because they express more genes. While this already suggests that fibroblasts are

more heterogeneous than mural cells, the question remains if the relative homogeneity of the mural cells in the UMAP landscape is due to masking through the presence in the same landscape of the more variable fibroblasts. However, this does not seem to be the case because:

1. Separate analysis of mural cells (see Figure 7 and Supplementary Figure 9) shows a similar (limited) dispersion of the organ-specific cell populations as when the mural cells are clustered together with the fibroblasts (see Figure 2a, b); thus, the presence of fibroblasts in the clustering and UMAP display does not appear to mask heterogeneity among the mural cells.
2. Clustering based on the limited gene sets representing different GO entities (see Figure 2c and Supplementary Figure 3c-e) shows less mural cell UMAP dispersal compared to fibroblasts irrespective of chosen GO-gene set. Again, fibroblasts expressed a higher (cumulative) number of genes than mural cells also here, e.g. 780 vs. 619 for ECM+matrisome, 1104 vs. 900 for cell-cell signaling, 1899 vs. 1635 for cell surface receptor signaling.
3. We nevertheless re-assessed dispersal following data down-sampling. There are several ways to do this. For simplicity, we removed all fibroblasts with more than 4000 expressed genes and re-calculated the UMAP dispersion for the remaining set of cells (see the Figure 1 and Legends in the appended Data for reviewer). This is a “conservative” way to down-sample, as it actually leads to a lower average number of genes per cell for the included fibroblasts than for the mural cells. In spite of this, the result showed a similar UMAP dispersion as the complete dataset including lung and brain (Figure 8), with organ origin as the main driver of dispersion of fibroblasts, whereas mural cells continue to cluster more homogeneously.

2. Fig. 3f: The authors show that gene expression is zoned for perisymal cells. It is not clear, however, alongside which axis and in which direction (as done in Fig. S7b))

We thank the reviewer for asking this question, alerting us to be more cautious in our use of the term *zonation*, which as the reviewer notes infers gene expression changes along an anatomical axis or direction. Because we have not yet defined any spatial point of reference (e.g. muscle surface, vessel/nerve bundle or other) or anatomical axis (e.g. A-P, D-V etc.), to which the perimysial heterogeneity can be related, we now call it *molecular diversity* instead of *zonation*. We briefly mention (see page 7, line 196-200) that this is an interesting question for future work, and have changed the terminology also in Figure 3 and Supplementary data 7.

3. Fig. 4 displays fibroblast heterogeneity in the heart and cross-references a Wif1 expressing subpopulation (that shows strong similarity with valve interstitial cell types that have previously been published) with transcriptionally similar cells from the skeletal muscle (perimysial cells). While this clearly illustrates the power of multi-organ high-resolution atlases, it is a little strange that the other subpopulations are not at all characterised or even mentioned in the text.

We thank the reviewer for pointing this out. The remaining fibroblast populations in the heart exhibited only limited dispersion in our UMAP analysis (Suppl. Figure 2a), which is in line with other published work (Farbehi et al., eLife 2019), but we realize that this notion did not come across very well. We have therefore rewritten this part in the revised version (the end of the *Results - Heart section*, page 8, line 242-244).

4. The authors elegantly show that fibroblast gene expression is zoned in a similar manner as for epithelial cells in the colonic mucosa. While this is a very interesting finding, it also makes room for new questions, e.g. is the zonation complementary or additive between cell types?, which cell type is instructive for establishing zonation?, what are the key molecules for establishing zonation? In-depth analyses might be beyond the scope for the manuscript, but the

authors could still hypothesize and discuss a bit more how they envision the zonation to be established.

We agree with the reviewer that the zonation of fibroblasts in the intestinal mucosa is intriguing. We followed the suggestion of the reviewer and have extended the discussion of these data with respect to the interplay and reciprocal signaling of intestinal (colonic) epithelial cells and fibroblasts, in the *Discussion section* (page 16, line 497-506).

5. The authors obtained only a very low number of pericytes from skeletal muscle or bladder and argue that this might be due to inefficient tissue dissociation. This explanation, as it stands, does not seem credible. As SMART-Seq2 is a plate-based scRNA-Seq platform and the authors used FACS sorting to enrich and purify cells, this would mean that the authors were only able to sort 10 cells from the whole organ. Is that correct or did the authors sort a larger number of cells and library generation was only successful for 10 cells? In the latter case, could the authors comment on what they think is the issue generating libraries from muscle/bladder pericytes as compared to other organs (RNA content, sensitivity)?

We appreciate this comment, which made us realize that we had not done a very good job in explaining that the transgenic reporter lines used to sort mural cells also captures fibroblasts, which are far more abundant. Thus, out of a large number of sorted cells (from e.g. the *Pdgfrb*^{GFP} mice) only 10 eventually turned out to be skeletal muscle and bladder pericytes. In other words, we sorted many cells from the skeletal muscle and bladder that exhibited similar features (FACS) as pericytes from the heart, however, their transcriptome eventually revealed that these cells were fibroblasts. We have rewritten the description to better explain this (page 11, line 334-336, 346-348).

Why do we capture so few pericytes from skeletal muscle, when by morphology we can readily find them in the tissue? We know from our previous experience and from others' published data (where pericytes are notoriously endothelium-contaminated) that it is hard to separate endothelial cells from pericytes during tissue disintegration. Therefore, we used anti-CD31 antibody to eliminate endothelial-mural doubles. In all our FACS runs, we could observe *contaminated* (e.g. *Pdgfrb*^{GFP}+ and CD31+) cells (as seen in Supplementary Figure 1a) to a varying extent. Note that these are not necessarily complete cell doublets, which would be possible to distinguish by size or DNA content, but mostly cell fragment contamination, as judged by the variation in amount (as little as 5% of endothelial transcriptome can be readily detected). To assure that we analyzed pure pericytes, we excluded all such double-positive cells by FACS and during the initial QC of the transcriptome data. This is now also more clearly described in the revised version (page 29, line 892-893, page 30, line 939-941). In conclusion, we therefore do not believe that the low number of pericytes from the skeletal muscle and bladder is due to difficulties in library preparation (low quality RNA), but rather the inherent difficulty in obtaining pure (free from endothelial fragment contamination) pericytes, especially from the skeletal muscle.

In the case of the bladder, as also discussed in the manuscript, we found that the bladder mucosa capillaries differ in their morphology compared to the other capillary beds. We also found that mural cells of the bladder mucosal capillaries exhibited a higher level of *Acta2* positivity. Therefore, it is not unlikely that bladder mural cells numbers are dominated by smooth muscle-like cells, simply because typical capillaries (i.e. ≈5 μm diameter microvessels) and typical pericytes are rare. We have included a brief mentioning of this in the revised version (page 11, line 346-348 and page 15, line 449-451).

6. The authors show in a very elegant approach that the matrisome is what drives fibroblast

heterogeneity. As total transcription factors expression did not seem to contribute much to the observed heterogeneity, the authors could use a reverse approach by running motif enrichment analyses for the differentially expressed matrix modulators in order to identify key transcription factors that drive heterogeneity (it is clear that the signature of a handful of differentially expressed transcription factors will be “diluted” within the bulk set). Furthermore, it would add to the understanding of pericyte heterogeneity if a similar analysis was performed.

This is an interesting point, which we admit had escaped our attention. We have now performed this analysis. As input gene list, we selected those genes of the ECM+matrisome gene-set that were also amongst the differentially expressed genes shown in Supplementary Figure 3b. This list of 155 genes was analyzed using the R-package *RcisTarget*, and the result is summarized in Table for reviewer and Data for reviewer Figure 2. As can be seen in the Data for reviewer Figure 2, the respective transcription factors to the identified motifs do not exhibit a pronounced cluster, or tissue-specific expression pattern. Overall, the expression of transcription factors can be considered as low, and more in-depth analysis would be required to draw reliable conclusions. Therefore, we interpret these new results as preliminary, and feel that the needed in-depth analysis to pinpoint potential transcription factors controlling ECM+matrisome gene-set expression lies beyond the scope for this manuscript.

7. Fig. 8: The authors integrate their current work with a previously published dataset on lung and brain fibroblasts/pericytes. Clustering revealed that lung and brain pericytes cluster into separate subpopulations, whereas the organs analyzed in this work rather fall into one cluster. Is this due to batch-effects or does it mirror fundamental differences in biology and function?

We are confident that the differences seen in the UMAP landscape of the integrated data (Figure 8) are not caused by batch-effects. First, the dispersion of the brain pericytes is similar to that observed for heart pericytes (relative to the main SMC cluster), and those two pericyte populations do exhibit substantial molecular differences (Figure 8d), i.e. qualitative differences in the expression of individual genes, exemplified by the ATP13a5 transporter which is highly and specifically expressed by brain pericytes (see also Vanlandewijck et al, Nature 2018); these differences cannot be caused by batch effects.

8. The authors used 24 individual mice. Fig S1c is supposed to clarify which cells in the cluster were derived from which mouse. Nevertheless, it is impossible to distinguish 24 distinct colors in the UMAP. The authors should include a table depicting how many cells came from which mouse per cluster.

We appreciate this comment, and agree that the use of 24 distinct colors was not optimal. Thus, we include a table depicting the absolute numbers of cells from each biological sample per cluster of the overall analysis (new Supplementary Table 5).

9. It is at times difficult to follow which pagoda cluster annotations were used in the Figures (e.g Fig. S2a, which I assume derives from Fig. 1).

We thank the reviewer for this remark, and have rephrased the Figure legends, in order to improve clarity around the cluster annotation in pagoda2.

Reviewer #4 (Remarks to the Author):

The authors have addressed my concerns and this work represents a useful resource for the field.

We thank the reviewer for the positive remarks.

Reviewer #5 (Remarks to the Author):

In general the response to reviewers were positive on many key problems in the original version, probably most importantly dealing with sample size. This certainly provides added rigor to an already very detailed and technically superb manuscript. Thus, the paper is improved.

We thank the reviewer for the positive comments.

However, a residual question by two reviewers were not answered. Essentially, with vast number of manuscripts with scRNAseq data, many focused on fibroblasts and mural cells already, how does this manuscript stand apart from the others? Two of the reviewers suggested functional studies or differences in disease states. Fibroblasts are key cell types in many disease states, and as such, their activation is probably their most important criteria. The authors did not add any data to this major critique and instead replied their data set will be important for fibroblast studies in general and differentiating from mural cells. This is true, but without functional data, or the potential changes in a disease state, the work remains descriptive.

Another lingering component that adds to the descriptive “feel” of the manuscript is there is no single marker to identify fibroblasts across organ systems. This is not the fault of the authors for the data, but it is likely something could have been done to analyze this further; e.g., in a particular disease states, does a common marker for fibroblasts appear? Adding more organs to the analysis didn’t necessarily strengthen the initial results, it just added another level of complexity to their story (for better or for worse).

We thank the reviewer for the careful evaluation and appreciation of our work. We do agree that functional analysis is of paramount interest to expand the analysis of fibroblasts and mural cells into various disease states. However, to fully address these aspects would be a rather time-consuming exercise, and we follow the Editor’s advice that this is therefore better suited for future studies.

We agree that our study is descriptive, and would like to stress that this was indeed the scope from the onset of the study, as such a comprehensive cross-organ analysis was lacking in the literature. We wanted to achieve the best possible description of the differentiated stage during adult homeostasis, to produce a platform based on which we and others may now initiate studies focused on specific pathological situations, such as fibrosis.

We believe the fact that there is no single marker uniquely identifying fibroblasts or mural cells *is* one of the very important findings in this study. With the help of our gene-sets for classification of fibroblasts and mural cells, we and others have a better chance to more objectively classify and annotate important cell populations in future studies.

In conclusion, we thank the reviewers for fair and insightful comments, which have made it possible for us to produce a stronger version of the manuscript.

Data for reviewer Figure 1

a Histogram of expressed genes per cell

a, Histogram of expressed genes per cell of all cells in the dataset (left), split into fibroblasts (clusters 1-5, 7-10, 12, 13, 15, grey) and mural cells (clusters 6, 11, 14, 16, brown), or the down-sampled dataset (right) where all fibroblasts with more than 4000 expressed genes (> 4000) were taken out. The median value of expressed genes is given in the histogram of the complete dataset.

b UMAP dimensional reduction

b, UMAP dispersion plot for the analysed down-sampled dataset, colour coded for the organ of origin. Mural cells, core smooth muscle cells (core SMC), pericytes as well as cardio-pulmonary SMC and interstitial SMC are indicated. Overall, the dispersion of mural cells is similar as observed for the analysis of the complete dataset (see Figure 2a, b for reference). The distribution of the remaining fibroblasts also recapitulated the results observed for the complete dataset; clustering in an organ-specific manner.

c UMAP dimensional reduction

c, UMAP dispersion plots showing the expression level of single genes used for annotation of the cell clouds in the UMAP plot shown in b.

Data for reviewer Figure 2

a Transcription factors found by motif analysis

a, Heat map of selected transcription factors identified by motif analysis of 155 differentially expressed genes from extracellular matrix + matrisome gene set (compare to Figure 2c), see Table for reviewer for input gene list and output results from *RcisTarget* R-package analysis. The two transcription factors (*Mbtps2*, *Arid3a*) with the highest NES (Normalized Enrichment Score) are indicated by black arrows and additional transcription factors (with multiple annotations) are highlighted with grey arrows.

b

b, Bar plots of the complete dataset showing the gene expression profiles of selected and indicated (a) transcription factors.

REVIEWERS' COMMENTS:

Reviewer #2 (Remarks to the Author):

The authors need to be congratulated for a meticulously executed and transparently communicated revision of their manuscript. It's a solid and complete story that will make an impactful and sustainable contribution to the literature.

RESPONSE TO REVIEWERS' COMMENTS:

Reviewer #2 (Remarks to the Author):

The authors need to be congratulated for a meticulously executed and transparently communicated revision of their manuscript. It's a solid and complete story that will make an impactful and sustainable contribution to the literature.

We thank the reviewer for the positive response and the constructive review process that helped to improve our manuscript.